# Type 2 and interferon inflammation regulate SARS-CoV-2 entry factor expression in the airway epithelium

Satria P. Sajuthi[1,16], Peter DeFord [1,16], Yingchun Li[1], Nathan D. Jackson[1], Michael T. Montgomery [1], Jamie L. Everman [1], Cydney L. Rios[1], Elmar Pruesse[1], James D. Nolin [1], Elizabeth G. Plender[1], Michael E. Wechsler[2], Angel C. Y. Mak [3], Celeste Eng[3], Sandra Salazar[3], Vivian Medina[4], Eric M. Wohlford[3,5], Scott Huntsman[3], Deborah A. Nickerson[6,7,8], Soren Germer[9], Michael C. Zody[9], Gonçalo Abecasis [10], Hyun Min Kang[10], Kenneth M. Rice [11], Rajesh Kumar[12], Sam Oh [3], Jose Rodriguez-Santana[4], Esteban G. Burchard[3,13] & Max A. Seibold [1,14,15✉]

Coronavirus disease 2019 (COVID-19) is caused by SARS-CoV-2, an emerging virus that utilizes host proteins ACE2 and TMPRSS2 as entry factors. Understanding the factors affecting the pattern and levels of expression of these genes is important for deeper understanding of SARS-CoV-2 tropism and pathogenesis. Here we explore the role of genetics and co-expression networks in regulating these genes in the airway, through the analysis of nasal airway transcriptome data from 695 children. We identify expression quantitative trait loci for both *ACE2* and *TMPRSS2*, that vary in frequency across world populations. We find *TMPRSS2* is part of a mucus secretory network, highly upregulated by type 2 (T2) inflammation through the action of interleukin-13, and that the interferon response to respiratory viruses highly upregulates *ACE2* expression. IL-13 and virus infection mediated effects on *ACE2* expression were also observed at the protein level in the airway epithelium. Finally, we define airway responses to common coronavirus infections in children, finding that these infections generate host responses similar to other viral species, including upregulation of *IL6* and *ACE2*. Our results reveal possible mechanisms influencing SARS-CoV-2 infectivity and COVID-19 clinical outcomes.

[1] Center for Genes, Environment, and Health, National Jewish Health, Denver, CO, USA. [2] Department of Medicine, National Jewish Health, Denver, CO, USA. [3] Department of Medicine, University of California San Francisco, San Francisco, CA, USA. [4] Centro de Neumología Pediátrica, San Juan, Puerto Rico. [5] Division of Pediatric Allergy and Immunology, University of California San Francisco, San Francisco, CA, USA. [6] Department of Genome Sciences, University of Washington, Seattle, WA, USA. [7] Northwest Genomics Center, Seattle, WA, USA. [8] Brotman Baty Institute, Seattle, WA, USA. [9] New York Genome Center, New York City, NY, USA. [10] Center for Statistical Genetics, University of Michigan, Ann Arbor, MI, USA. [11] Department of Biostatistics, University of Washington, Seattle, WA, USA. [12] Department of Pediatrics, Ann and Robert H. Lurie Children's Hospital of Chicago, Northwestern University, Chicago, IL, USA. [13] Department of Bioengineering and Therapeutic Sciences, University of California San Francisco, San Francisco, CA, USA. [14] Department of Pediatrics, National Jewish Health, Denver, CO, USA. [15] Division of Pulmonary Sciences and Critical Care Medicine, University of Colorado-AMC, Aurora, CO, USA. [16]These authors contributed equally: Satria P. Sajuthi, Peter DeFord. ✉email: seiboldm@njhealth.org

In December of 2019, a novel Coronavirus, SARS-CoV-2, emerged in China and has gone on to trigger a global pandemic of Coronavirus Disease 2019 (COVID-19), the respiratory illness caused by this virus[1]. While most individuals with COVID-19 experience mild cold symptoms (cough and fever), some develop more severe disease including pneumonia, which often necessitates mechanical ventilation[2]. In fact, an estimated 5.7% of COVID-19 illnesses are fatal[3]. Enhanced risk of poor outcomes for COVID-19 has been associated with a number of factors including advanced age, male sex, and underlying cardiovascular and respiratory conditions[4,5]. Yet, while the majority of serious COVID-19 illness occurs in adults over 60, children are also susceptible to infection, with the Centers for Disease Control and Prevention (CDC) reporting that 2% of confirmed cases being in patients under 18 years of age. Of these COVID-19 cases in children, the CDC estimates that 5.7–20% are hospitalized and 0.58–2.0% require ICU stays[6]. Moreover, recent data from China suggests that 38% of COVID-19 cases occurring in children are of moderate severity and 5.8% are severe or critical[7], although not all cases were confirmed in this data set. In addition, a small minority of SARS-CoV-2 infections in children have been associated with a severe inflammatory syndrome similar to Kawasaki's disease. Together these data highlight a need to study risk factors of COVID-19 illnesses in children.

One factor that may underlie variation in clinical outcomes of COVID-19 is the extent of gene expression in the airway of the SARS-CoV-2 entry receptor, ACE2, and TMPRSS2, the host protease that cleaves the viral spike protein and thus allows for efficient virus-receptor binding[8]. Expression of these genes and their associated programs in the nasal airway epithelium is of particular interest given that the nasal epithelium is the primary site of infection for upper airway respiratory viruses, including coronaviruses, and acts as the gateway through which upper airway infections can spread into the lung. The airway epithelium is composed of multiple resident cell types (e.g., mucus secretory, ciliated, basal stem cells and rare epithelial cell types) interdigitated with immune cells (e.g., T cells, mast cells, macrophages), and the relative abundance of these cell types in the epithelium can greatly influence the expression of particular genes[9–11], including ACE2 and TMPRSS2. Furthermore, since the airway epithelium acts as a sentinel for the entire respiratory system, its cellular composition, along with its transcriptional and functional characteristics, are significantly shaped by interaction with environmental stimuli. These stimuli may be inhaled (e.g., cigarette smoke, allergens, microorganisms) or endogenous, such as when signaling molecules are produced by airway immune cells present during different disease states. One such disease state is allergic airway inflammation caused by type 2 (T2) cytokines (IL-4, IL-5, IL-13), which is common in both children and adults and has been associated with the development of both asthma and chronic obstructive pulmonary disease (COPD) in a subgroup of patients[12–14]. T2 cytokines are known to greatly modify gene expression in the airway epithelium, both through transcriptional changes within cells and epithelial remodeling in the form of mucus metaplasia[12,15,16]. Microbial infection is another strong regulator of airway epithelial expression. In particular, respiratory viruses can modulate the expression of thousands of genes within epithelial cells, while also recruiting and activating an assortment of immune cells[17–19]. Even asymptomatic nasal carriage of respiratory viruses, which is especially common in childhood, has been shown to be associated with both genome-wide transcriptional re-programming and infiltration of macrophages and neutrophils in the airway epithelium[20], demonstrating how viral infection can drive pathology even without overt signs of illness.

Genetic variation is another factor that may regulate gene expression in the airway epithelium. Indeed, expression quantitative trait loci (eQTL) analyses carried out in many tissues have suggested that the expression of as many as 70% of genes are influenced by genetic regulatory variants[21]. The severity of human rhinovirus (HRV) respiratory illness has specifically been associated with genetic variation in the epithelial genes CDHR3[22] and the ORMDL3[23] and, given differences in genetic variation across world populations, it is possible that functional genetic variants in SARS-CoV-2-related genes could partly explain population differences in COVID-19 clinical outcomes.

Finally, there are important questions regarding the host response to SARS-CoV-2 infection. For example, it is unclear whether specific antiviral defenses in the epithelium are blocked by SARS-CoV-2 or whether the virus may trigger epithelial or immune cell pathways that prolong airway infection, and/or even incite a hyperinflammatory state in the lungs in some individuals that leads to more severe disease. Although large cohorts of subjects infected by the novel coronavirus are still lacking, much can be learned by exploring transcriptional responses to other coronavirus strains. In particular, because nasal airway brushings capture both epithelial and immune cells present at the airway surface, such samples collected from a cohort of subjects infected by a range of viruses provide an opportunity to comprehensively investigate the potentially varied and cascading effects of coronavirus infection on airway expression and function.

In this study, we first use single-cell RNA-sequencing (scRNA-seq) to elucidate the cellular distribution of ACE2 and TMPRSS2 expression in the nasal airway epithelium. We also perform network and eQTL analysis of bulk gene expression data on nasal airway epithelial brushings collected from a large cohort of asthmatic and healthy children in order to identify the genetic and biological regulatory mechanisms governing ACE2 and TMPRSS2 expression. We then validate the effects on ACE2 expression at the protein level in the airway epithelium. We then use multi-variable modeling to estimate the relative contribution of these factors to population variation in the expression of these two genes, and by performing experiments on mucociliary airway epithelial cultures confirm a dominant role for both T2 inflammation and viral infection in regulating the expression of ACE2 and TMPRSS2. Finally, we define the cellular and transcriptional responses to in vivo coronavirus infections in the nasal airway of children.

## Results

**Cell-type specificity of ACE2 and TMPRSS2 in the nasal airway.** We first examined ACE2 and TMPRSS2 expression at a cell-type level through single-cell RNA-sequencing (scRNA-seq) of a nasal airway epithelial brushing from an adult, asthmatic subject. Shared Nearest Neighbor (SNN)-based clustering of 8,291 cells identified nine epithelial and three immune cell populations (Fig. 1a and Supplementary Data 1). We found that seven epithelial cell populations contained ACE2+ cells (at low frequency), with the highest frequency of positive cells found among basal/early secretory cells, ciliated cells, and secretory cells (Fig. 1b). We did not observe meaningful ACE2 expression among any of the immune cell populations, which included T cells, dendritic cells, and mast cells. We found TMPRSS2 to be expressed by all epithelial cell types, with a higher frequency of positive cells among the different cell types, compared to ACE2 (Fig. 1b, c). A small number of mast cells were also TMPRSS2+ (Fig. 1c).

**TMPRSS2 is a T2 inflammation-induced mucus secretory gene.** We next sought to determine the variation in nasal epithelial expression of ACE2 and TMPRSS2 across healthy and asthmatic

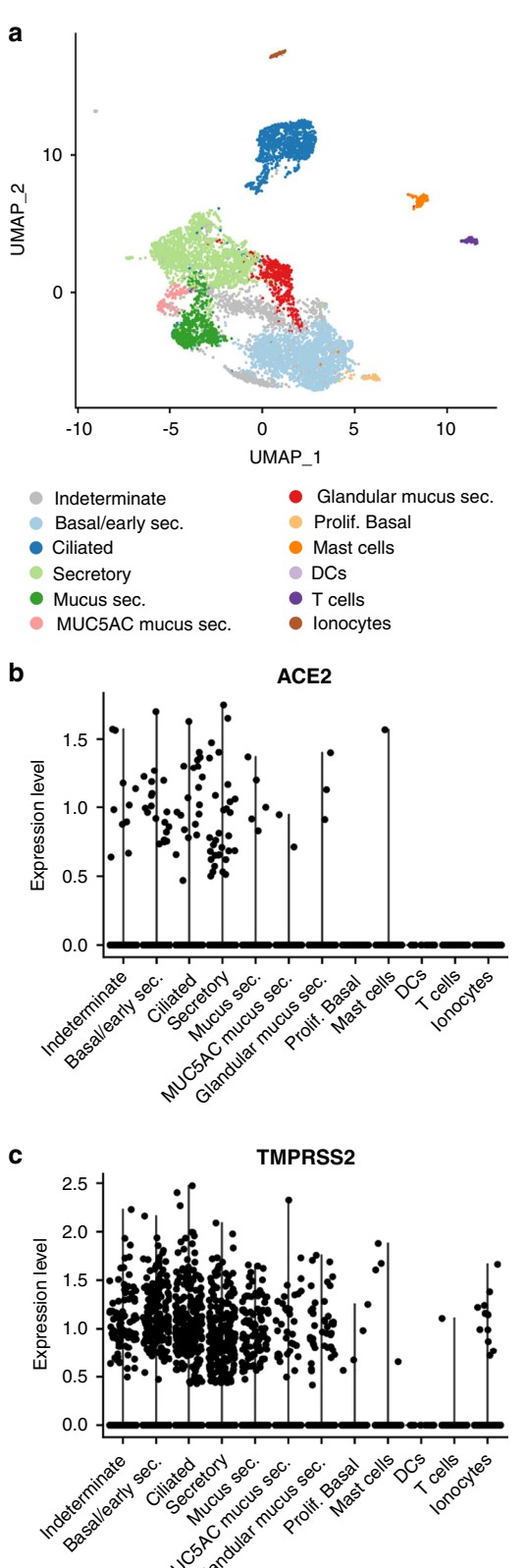

**Fig. 1 ACE2 and TMPRSS2 are expressed by multiple nasal airway cell types. a** UMAP visualization of 8,291 cells derived from a human nasal airway epithelial brushing depicts multiple epithelial and immune cell types identified through unsupervised clustering. **b** Log Count Per Million (CPM) normalized expression of ACE2 in epithelial and immune cell types. **c** Log CPM normalized expression of TMPRSS2 in epithelial and immune cell types.

children and to identify biological mechanisms that regulate this variation. Thus, we performed weighted gene co-expression network analysis (WGCNA) on whole-transcriptome sequencing data from nasal airway brushings of 695 Puerto Rican healthy and asthmatic children in the Genes-Environments and Asthma in Latino Americans II study (GALA II). This analysis identified 54 co-expression networks representing cell-type-specific expression programs such as ciliogenesis, mucus secretion, and pathways of immunity and airway inflammation (Supplementary Data 2). The TMPRSS2 gene was contained within one of a set of three highly correlated networks exhibiting strong enrichments for mucus secretory cell genes and pathways (Fig. 2a and Supplementary Data 2 and 3). For example, the black network, which was highly correlated with TMPRSS2 expression ($r = 0.64$, $p = 1e-82$), was strongly enriched for Asparagine N-linked glycosylation (R-HSA-446203) and COPI-mediated anterograde transport (R-HSA-6807878) pathways. These and other enriched pathways are involved in the normal processing and transport of mucin proteins (Fig. 2a). TMPRSS2 itself fell within and was highly correlated with expression of the pink network ($r = 0.68$, $p = 3e-97$), which was highly enriched for mucus goblet cell markers ($p = 2e-6$, Fig. 2a, b). The pink network was also enriched for genes involved in the O-linked glycosylation of mucins (R-HSA-913709) pathway ($p = 9e-4$), which is vital to the function of mucins, especially those produced by mucus secretory cells induced by T2 inflammation ($r = 0.68$, $p = 3e-97$, Fig. 2a, b). In fact, we found that this network contained the T2 cytokine IL13 while being particularly enriched for genes known to mark and transcriptionally regulate IL-13-induced mucus metaplasia (FCGBP, SPDEF, FOXA3). The saddle brown network was also related to mucus secretory cells, and contained the most canonical T2 inflammation markers[12,24] including POSTN, CLCA1, CPA3, IL1RL1, CCL26, and was strongly correlated with both TMPRSS2 ($r = 0.61$, $p = 5e-72$, Fig. 2c) and the other T2 mucus secretory network (pink) ($r = 0.92$, $p = 3e-280$, Supplementary Data 4). In contrast, we found ACE2 expression to be strongly negatively correlated with expression of both T2 networks (pink: $r = -0.61$, $p = 3e-72$, saddle brown: $r = -0.7$, $p = 2e-102$, Fig. 2e, f). To identify subjects with high and low T2 inflammation, we hierarchically clustered all subjects based on the expression of genes in the canonical T2 network (saddle brown). This resulted in the identification of two distinct groups we labeled as T2-high ($n = 364$) and T2-low ($n = 331$) (Supplementary Fig. 1a). We found that this expression-derived T2 status was strongly associated with traits known to be driven by T2 inflammation including immunoglobulin E (IgE) levels, exhaled nitric oxide (FeNO), blood eosinophils, and asthma diagnosis (Supplementary Fig. 1b–e). Notably, TMPRSS2 levels were 1.3-fold higher in T2-high subjects ($p = 1e-62$), while, ACE2 expression was 1.4-fold lower in T2-high subjects ($p = 2e-48$) (Fig. 2d, g).

To investigate whether the strong in vivo relationship between airway T2 inflammation and TMPRSS2/ACE2 expression is causal in nature, we performed in vitro stimulation of paired air–liquid interface (ALI) mucociliary airway epithelial cultures with 72 h of IL-13, a known regulator of type 2 inflammation[25,26], or mock stimulus ($n = 5$ donors, Fig. 3a). Performing paired differential expression analysis between the mock and IL-13 stimulated cultures, we found that ACE2 and TMPRSS2 were strongly downregulated and upregulated, respectively, supporting our in vivo analysis results ($\log_2$ fold change [$\log_2$FC] $= -0.67$, $p = 5e-3$, $\log_2$FC $= 1.20$, $p = 5e-9$, Fig. 3b, c). To better understand the cellular basis of TMPRSS2 and ACE2 regulation by IL-13, we leveraged scRNA-seq data previously generated on tracheal airway epithelial cultures that were chronically stimulated (10 days) with IL-13 or control media (Fig. 3a, d). Similar to

**a**

| Network (color) | Network size (#genes) | Select genes | Pathway enrichment (*p-adj*) | Cell-type enrichment (*p-adj*) |
|---|---|---|---|---|
| Mucus secretory (black) | 477 | COPA COPB2 COPG1 CREB3L1 XBP1 | Asparagine N-linked glycosylation (*2e–14*)<br>Transport to the Golgi and subsequent modification (*7e–10*)<br>Interleukin-13 (IL-13) human airway epithelial cells (*4e–9*)<br>COPI-mediated anterograde transport (*9e–6*) | Goblet (*2e–17*)<br>Diff. basal (*0.03*) |
| T2mucus secretory (pink) | 446 | SPDEF FCGBP FOXA3 IL13 BPIFB1 | Interleukin-13 (IL-13) human airway epithelial cells (*3e–47*)<br>Mucin type O-glycan biosynthesis (*9e–5*)<br>O-linked glycosylation of mucins (*3e–4*) | Goblet (*2e–6*)<br>Serous (*3e–3*) |
| CanonicalT2 biomarker (saddlebrown) | 156 | CLCA1 CCL26 POSTN IL1RL1 CPA3 | Interleukin-13 (IL-13) human airway epithelial cells (*9e–29*)<br>Interleukin-4 human keratinocyte (*0.02*) | Mast cell/basophil type 1 (*6e–11*)<br>Mast cell/basophil type 2 (*6e–11*) |

**b** $r = 0.68, p = 3e–97$

**c** $r = 0.61, p = 5e–72$

**d** $log_2FC = 0.41, p = 1e–62$

**e** $r = -0.61, p = 3e–72$

**f** $r = -0.7, p = 2e–102$

**g** $log_2FC = -0.53, p = 2e–48$

**Fig. 2 *TMPRSS2* is part of a T2 inflammation-induced mucus secretory network. a** WGCNA identified networks of co-regulated genes related to mucus secretory function (black), T2 inflammation-induced mucus secretory function (pink), and canonical T2 inflammation biomarkers (saddle brown). *TMPRSS2* was within the pink network. Select pathway and cell-type enrichments for network genes are shown. Enrichment *p*-values were obtained from a one-sided hypergeometric test. Benjamini–Hochberg correction was used to control for false discovery rate. **b** Scatterplot revealing a strong positive correlation between *TMPRSS2* expression and summary (eigengene) expression of the T2 inflammatory, mucus secretory network. *p*-values were obtained from the two-sided Pearson correlation test. **c** Scatterplot revealing a strong positive correlation between *TMPRSS2* expression and summary (eigengene) expression of the canonical T2 inflammation biomarker network. *p*-values were obtained from the two-sided Pearson correlation test. **d** Box plots revealing strong upregulation of *TMPRSS2* expression among T2-high (*n* = 364) compared to T2-low (*n* = 331) subjects. Differential expression testing between T2-high and T2-low groups was performed using a two-sided Wilcoxon test. **e** Scatterplot revealing a strong negative correlation between *ACE2* expression and summary (eigengene) expression of the T2 inflammation mucus secretory network. *p*-values were obtained from the two-sided Pearson correlation test. **f** Scatterplot revealing a strong negative correlation between *ACE2* expression and summary (eigengene) expression of the canonical T2 inflammation biomarker network. *p*-values were obtained from the two-sided Pearson correlation test. **g** Box plots revealing strong downregulation of *ACE2* expression among T2-high (*n* = 364) compared to T2-low (*n* = 331) subjects. Differential expression testing between T2-high and T2-low groups was performed using a two-sided Wilcoxon test. Box centers give the median, upper and lower box bounds correspond to first and third quartiles and the upper/lower whiskers extend from the upper/lower bounds up to/down from the largest/smallest value, no further than 1.5 × IQR from the upper/lower bound (where IQR is the inter-quartile range). Data beyond the end of whiskers are plotted individually.

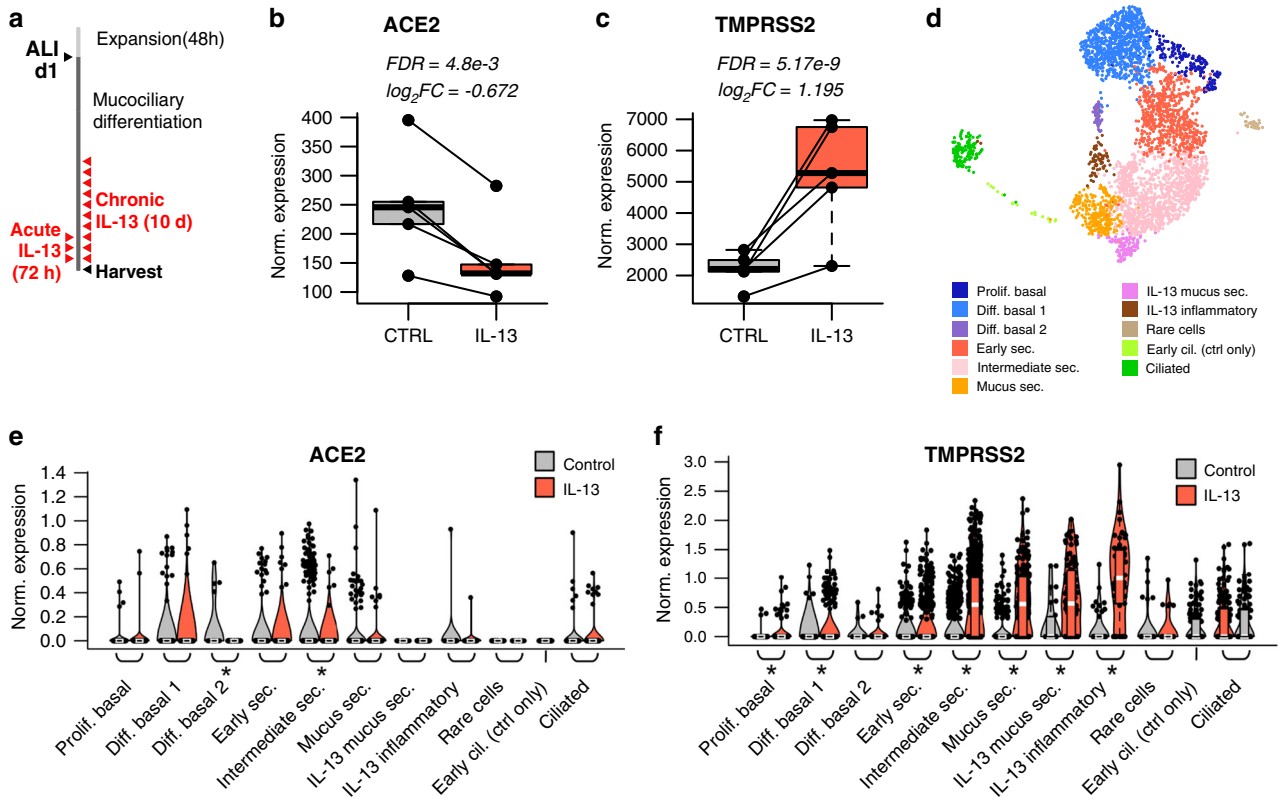

**Fig. 3 *ACE2* and *TMPRSS2* are both regulated by IL-13 in the airway epithelium. a** Experimental schematic detailing the timeline for differentiation of basal airway epithelial cells into a mucociliary airway epithelium and treatment with chronic (10 days) or acute (72 h) IL-13 (10 ng mL$^{-1}$). **b** Box plots of count-normalized expression between paired ($n = 5$ pairs) nasal airway cultures (control/IL-13) revealing strong downregulation of bulk *ACE2* expression with IL-13 treatment. Differential expression results are from DESeq2. Benjamini–Hochberg correction was used to control for false discovery rate. **c** Box plots of count-normalized expression between paired ($n = 5$ pairs) nasal airway cultures (control/IL-13) revealing strong upregulation of bulk *TMPRSS2* expression with IL-13 treatment. Differential expression results are from DESeq2. Benjamini–Hochberg correction was used to control for false discovery rate. **d** UMAP visualization of 6,969 cells derived from control and IL-13 stimulated tracheal airway ALI cultures depict multiple epithelial cell types identified through unsupervised clustering. **e** Violin plots of normalized *ACE2* expression across epithelial cell types from tracheal airway ALI cultures, stratified by treatment (gray = control, red = IL-13). Differential expression using a two-sided Wilcoxon test was performed between control and IL-13-stimulated cells with significant differences in expression for a cell type indicated by a * ($p < 0.05$). *p*-values (left to right) = 0.62; 0.18; 2.8e−4; 0.66; 4.6e−6; 0.08; NA; 0.77; NA; NA; 0.12. **f** Violin plots of normalized *TMPRSS2* expression across epithelial cell types from tracheal airway ALI cultures, stratified by treatment (gray = control, red = IL-13). Differential expression using a two-sided Wilcoxon test was performed between control and IL-13-stimulated cells with significant differences in expression for a cell type indicated by a * ($p < 0.05$). *p*-values (left to right) = 9.4e−4; 1.4e−11; 0.51; 2.5e−14; 6.9e−117; 1.5e−38; 2.3e−4; 2.3e−10; 0.46; NA; 0.95. Box centers give the median, upper and lower box bounds correspond to first and third quartiles and the upper/lower whiskers extend from the upper/lower bounds up to/down from the largest/smallest value, no further than 1.5 × IQR from the upper/lower bound (where IQR is the inter-quartile range). Data beyond the end of whiskers are plotted individually.

our results from in vivo nasal scRNA-seq data, we observed that *ACE2* expression was highest among basal, ciliated, and early/intermediate secretory cell populations, with *ACE2* being significantly downregulated by IL-13 among both basal and intermediate secretory cells (Fig. 3e). Also mirroring the in vivo scRNA-seq data, *TMPRSS2* was expressed across all epithelial cell types, but at a higher frequency among secretory cells (Fig. 3f). IL-13 stimulation-induced marked upregulation of *TMPRSS2* in early secretory, intermediate secretory, and mature mucus secretory cell populations (Fig. 3f). Furthermore, IL-13 stimulated mucus metaplasia that resulted in the development of a novel mucus secretory cell type and an IL-13 inflammatory epithelial cell that both highly expressed *TMPRSS2* (Fig. 3f). Together, our in vivo and in vitro analyses strongly suggest that *TMPRSS2* is part of a mucus secretory cell network that is highly induced by IL-13-mediated T2 inflammation.

**ACE2 is a virally induced interferon response network gene.** Returning to the in vivo nasal airway epithelial expression networks, we found that *ACE2* expression was highly correlated with the expression of two networks (purple and tan) (purple: $r = 0.74$, $p = 3e−120$, tan: $r = 0.72$, $p = 2e−110$, Fig. 4a, b). The purple network was highly enriched for genes that mark cytotoxic T cells and antigen-presenting dendritic cells, both of which are particularly abundant in a virally infected epithelium (Fig. 4c and Supplementary Data 2), whereas the tan network was strongly enriched for interferon and other epithelial viral response genes (*IFI6*, *IRF7*, *CXCL10*, *CXCL11*) (Fig. 4c and Supplementary Data 2). Clustering of subjects based on the interferon response network genes resulted in two groups, one highly (interferon-high = 78) and one lowly (interferon-low = 617) expressing these interferon response network genes (Supplementary Fig. 2). We found that *ACE2* expression was 1.7-fold higher in the interferon-high vs. interferon-low group (Fig. 4d). In a previous study, we found that children with nasal gene expression characteristic of the interferon network tended to be infected with a respiratory virus, despite being asymptomatic[20]. To explore the possibility of this relationship in our current data

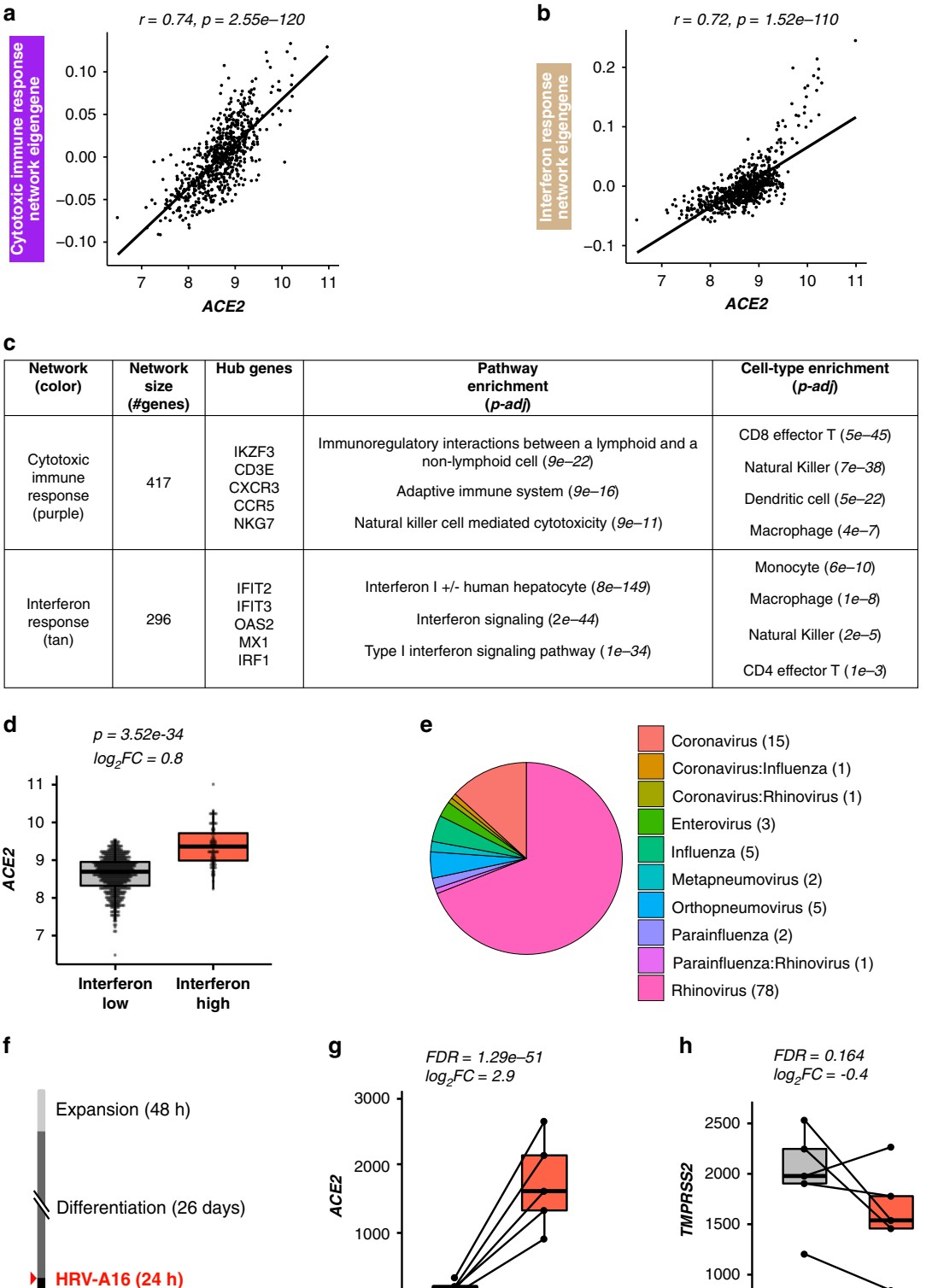

set, we metagenomically analyzed the RNA-seq data for all subjects to identify those harboring reads for a respiratory virus. This analysis found that 16% of GALA II children were asymptomatically harboring a respiratory virus from one of seven general respiratory virus groups (Fig. 4e). Strikingly, we found that 82% of interferon-high subjects were virus carriers compared to only 8% of interferon-low subjects (Supplementary Fig. 2B). These results demonstrate how asymptomatic virus

carriage nonetheless stimulates an active viral response that includes *ACE2*.

To directly test the effect of respiratory virus infection on epithelial *ACE2* gene expression, we again employed our ALI mucociliary epithelial culture system. Performing mock or human rhinovirus-A16 infection of mature cultures (day 27, Fig. 4f) from five donors, we found 7.7-fold upregulation of *ACE2* gene expression with HRV-A infection ($p = 1.3e-51$, Fig. 4g). In

**Fig. 4 *ACE2* is an interferon network gene regulated by a viral infection. a** Scatterplot revealing a strong positive correlation between *ACE2* expression and summary (eigengene) expression of the cytotoxic immune response network (purple). *p*-values were obtained from a two-sided Pearson correlation test. **b** Scatterplot revealing a strong positive correlation between *ACE2* expression and summary (eigengene) expression of the interferon response network (tan). *p*-values were obtained from a two-sided Pearson correlation test. **c** WGCNA analysis identified networks of co-regulated genes related to cytotoxic immune response (purple) and interferon response (tan). *ACE2* was within the purple network. Select pathway and cell-type enrichments for network genes are shown. Enrichment *p*-values were obtained from a one-sided hypergeometric test. Benjamini–Hochberg correction was used to control for false discovery rate. **d** Box plots of count-normalized expression from GALA II nasal epithelial samples reveal strong upregulation of *ACE2* expression among interferon-high ($n = 78$) compared to interferon-low ($n = 617$) subjects. Differential expression results are from DESeq2. Benjamini–Hochberg correction was used to control for false discovery rate. **e** Pie graph depicting the percentage of each type of respiratory virus infection found among GALA II subjects in whom viral reads were found. **f** Experimental schematic detailing a timeline for differentiation of basal airway epithelial cells into a mucociliary airway epithelium and experimental infection with HRV-A16. **g** Box plots of count-normalized expression between paired ($n = 5$ pairs) nasal airway cultures (control/HRV-A16 infected) revealing strong upregulation of *ACE2* expression with HRV-A16 infection. Differential expression results are from DESeq2. Benjamini–Hochberg correction was used to control for false discovery rate. **h** Box plots of count-normalized expression between paired ($n = 5$ pairs) nasal airway cultures (control/HRV-A16-infected) revealing no effect of HRVA-16 on *TMPRSS2* expression. Differential expression results are from DESeq2. Benjamini–Hochberg correction was used to control for false discovery rate. Box centers give the median, upper and lower box bounds correspond to first and third quartiles and the upper/lower whiskers extend from the upper/lower bounds up to/down from the largest/smallest value, no further than 1.5 × IQR from the upper/lower bound (where IQR is the inter-quartile range). Data beyond the end of whiskers are plotted individually.

contrast, we only observed a trend for the downregulation of *TMPRSS2* gene expression among virally infected subjects (Fig. 4h). These results confirm the strong regulation of *ACE2* gene expression by HRV-A viral infection and likely other respiratory viruses.

**ACE2 protein is regulated by T2 stimulus and viral infection.** We next evaluated whether the T2 and viral inflammation-driven modulation of *ACE2* observed at the mRNA level extends to the protein level as well. Specifically, we performed immunohistochemistry for ACE2 protein on replicate mature ALI mucociliary nasal airway epithelial cultures from a childhood asthmatic donor in the GALA II cohort. The replicate cultures were either mock-treated, IL-13 treated (5 days), or infected with HRV-A or HRV-C (24 h post-infection). In mock-treated cultures, we observed ACE2 protein staining both sporadically in the cytoplasm, and along the apical surface of epithelial cells, many times colocalizing with ciliated cells (Fig. 5a, f and Supplementary Fig. 3a). Due to its potential to mediate virus binding, we focused the quantification of ACE2 staining to that found on the apical surface. Five days of IL-13 treatment strongly downregulated ACE2 apical staining (2.4-fold, $p = 0.03$, Fig. 5b, e and Supplementary Fig. 3b), which was conversely strongly increased by both acute HRV-A (2.0-fold, $p = 2e{-}3$, Fig. 5c–e and Supplementary Fig. 3c) and HRV-C infection (1.7-fold, $p = 0.02$, Fig. 5e).

To explore more definitively the relationship between the presence of ciliated cells and ACE2 surface protein expression, we treated nasal basal airway epithelial cells throughout ALI mucociliary differentiation with either mock stimulus (Fig. 5f), IL-13 (blocks ciliated cell differentiation, Fig. 5g) or the γ-secretase inhibitor DAPT (stimulates ciliated cell differentiation, Fig. 5h). The IL-13 differentiated cultures exhibited the expected marked increase in mucus secretory cells with no ciliated cells present. This loss of ciliated cells among the IL-13 stimulated cultures was matched by an almost complete loss of apical ACE2 protein (41.0-fold, $p = 1e{-}3$, Fig. 5g, i). In contrast, DAPT treatment generated the expected full ciliation of the ALI cultures, which corresponded with a marked increase in apical ACE2 protein expression (2.0-fold, $p = 0.02$, Fig. 5h, i). We then evaluated histological sections of two human tracheas (Fig. 5j and Supplementary Fig. 3d–f), finding the most prominent ACE2 expression localized to the apical surface of the tracheal epithelium, colocalizing strongly with ciliated cells. Together, these results confirm at a protein level the downregulation and upregulation of ACE2 caused by IL-13 stimulation and

virus infection of the airway epithelium, respectively, while also revealing ACE2 protein at the apical epithelial surface, particularly in ciliated cells.

**Genetic regulation of *ACE2* and *TMPRSS2* in the nasal airway.** We next explored the role of genetic regulatory variants in the epithelial expression of *ACE2* and *TMPRSS2*. To do this, we performed *cis*-eQTL analysis for these two genes, using nasal gene expression and genome-wide genetic variation data collected from the GALA II study children. We identified 316 and 36 genetic variants significantly associated with the expression of *ACE2* and *TMPRSS2*, respectively (Fig. 6a, b). Stepwise forward–backward regression analysis of these eQTL variants revealed a single independent eQTL variant (rs181603331) for the *ACE2* gene ($p = 6e{-}23$), located ~20 kb downstream of the *ACE2* gene (Fig. 6a). This rare eQTL variant (allele frequency [AF] = 1%) was associated with a large decrease in *ACE2* expression ($\log_2 A_{FC} = -1.6$) (Fig. 6c).

A similar analysis of the *TMPRSS2* eQTL variants yielded three independent eQTL variants (rs1475908 AF = 20%, rs74659079 AF = 4%, and rs2838057 AF = 13%, Fig. 6b). The eQTL variant rs1475908 was associated with a decrease in *TMPRSS2* expression ($\log_2$ allelic fold change [$\log_2 A_{FC}$] = $-0.37$, Fig. 6d), whereas both the rs74659079 and rs2838057 eQTL variants were associated with increased *TMPRSS2* expression ($\log_2 A_{FC} = 0.38$, 0.43, respectively, Supplementary Fig. 4).

Considering that a large number of eQTL effects are very consistent across populations, we next examined the frequency of these variants among eight world populations listed in the gnomAD genetic variation database (v2.1.1). We found that the *ACE2* eQTL variant was only present in people of African descent and at a low frequency (AF = 0.7%, Fig. 6e). In contrast, the *TMPRSS2* eQTL variant associated with decreased expression, rs1475908, occurred across all world populations, with the highest allele frequencies among East Asians (AF = 38%), Europeans (AF = 35%), intermediate frequencies among Africans (AF = 26%) and Ashkenazi Jews (AF = 23%), and the lowest frequency among Latinos (AF = 17%). The two *TMPRSS2* eQTL variants associated with increased expression exhibited much more disparate allele frequencies across world populations. Namely, the allele frequency of rs74659079 is above 1% only among people of African descent (AF = 11%) and 4% in the participating Puerto Rican population. Likewise, the rs2838057 eQTL variant, which was associated with increased *TMPRSS2* expression was present at a frequency of 32% in East Asians, 20% in Latinos, and <10% in all other world populations.

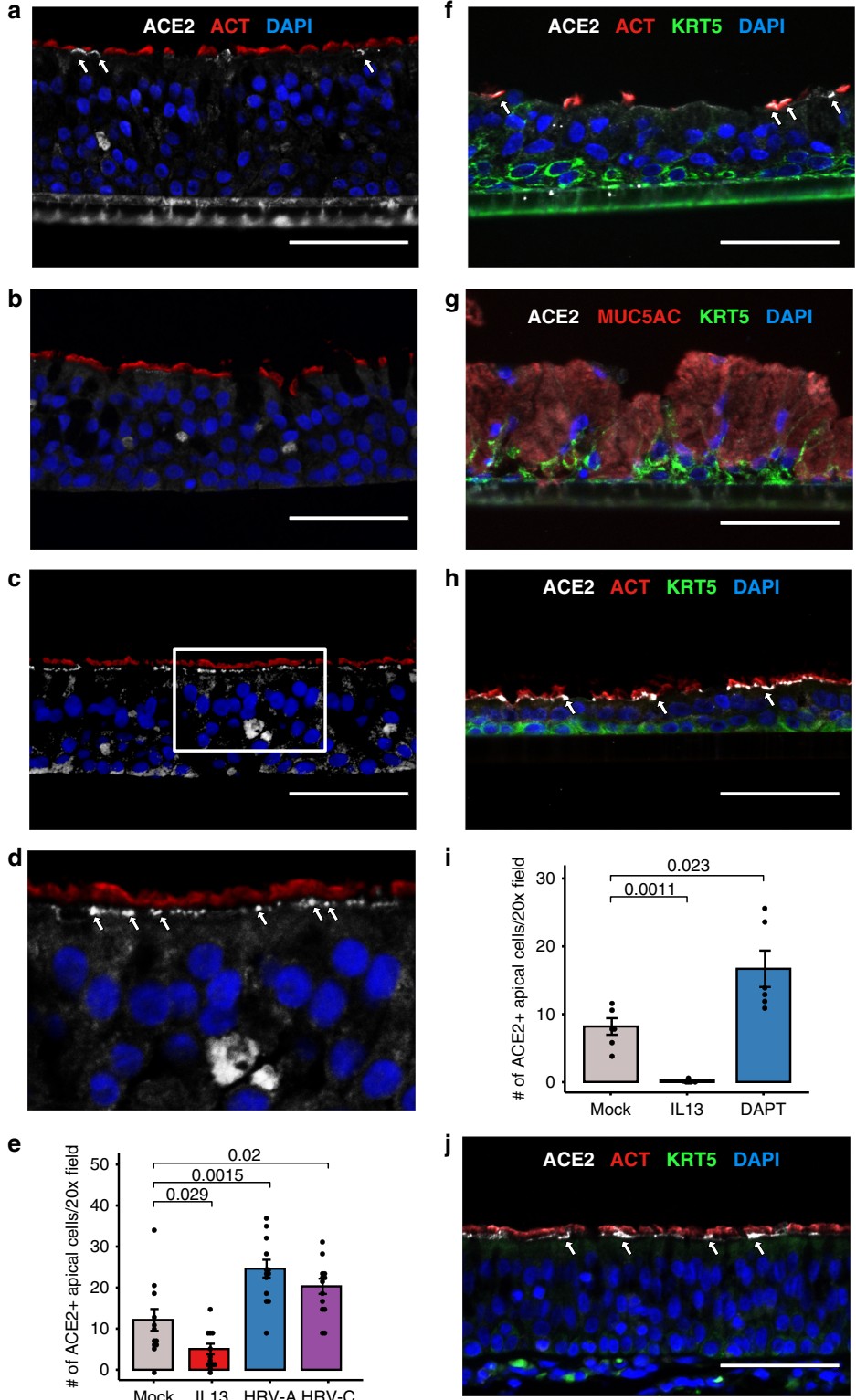

## Multi-variable modeling of airway *ACE2* and *TMPRSS2* levels.

Our analyses indicate that T2 inflammation, interferon/viral response signaling, and genetics are all determinants of *ACE2* and *TMPRSS2* gene expression in the airway epithelium of children. Therefore, we next sought to determine the relative importance of these factors in determining levels of these genes using multi-variable regression analysis. We included asthma status, age, and sex as model covariates since chronic lung disease, increasing age, and male sex have all been associated with increased risk of poor

COVID-19 illness outcomes. Modeling *ACE2* expression among GALA II children, we found that T2 and interferon statuses had the strongest effects on *ACE2* expression ($p = 1.6\text{e-}57$, $p = 6.5\text{e}-43$, respectively), with T2-low and interferon-high individuals exhibiting the highest levels of expression. These two variables independently explained 24% and 17% of the variance in *ACE2* expression (Table 1). While the *ACE2* eQTL variant, rs181603331, was associated with a notable decrease in *ACE2* levels, it only accounted for 1.2% of the variance, reflecting the

**Fig. 5 Airway surface ACE2 protein is regulated by IL-13 and viral infection. a–c** Immunofluorescence staining of in vitro nasal airway epithelial ALI cultures (derived from a single GALA II asthmatic child) derived from vehicle-treated (**a**), 5 days IL-13(10 ng mL$^{-1}$) treated (**b**), and 24 h HRV-A infected epithelium (**c**). Representative images of ciliated cells (ACT; red) and ACE2-positive (white) cells. The nuclei were counterstained with DAPI (blue). ACE2 protein was located in the apical compartment and decreased with IL-13 treatment and increased in HRV-A infection. Scale bars: 60 μm. **d** Zoomed in image of **c**. **e** Quantification of apical ACE2-positive cells per 20x field for each condition in the IL-13/HRV mature culture stimulation experiment (n = 1 donor). p-values were obtained from a two-sided t-test. Data are presented as mean values ± SEM. **f–h** Immunofluorescence staining of in vitro nasal airway epithelial ALI cultures from a single child non-asthmatic donor derived from vehicle-treated (**f**), 21 days IL-13 (10 ng mL$^{-1}$) treated (**g**), and 21 days DAPT(5 μM) treated epithelium (**h**). Representative images of ciliated cells (ACT; red), basal cells (KRT5; green), and ACE2-positive (white) cells. The nuclei were counterstained with DAPI (blue). ACE2 protein located in the apical compartment and decreased in IL-13 treatment and increased in DAPT treatment. Scale bars: 60 μm. **i** Quantification of apical ACE2-positive cells per 20x field for each condition in the IL-13/DAPT differentiation experiment (n = 1 donor). p-values were obtained from a two-sided t-test. Data are presented as mean values ± SEM. **j** Immunofluorescence staining of in vivo trachea tissues from a single healthy donor. Representative images of ciliated cells (ACT; red), basal cells (KRT5; green), and ACE2-positive (white) cells. The nuclei were counterstained with DAPI (blue). ACE2 protein located in the apical compartment of ACT positive ciliated cells. Scale bars: 60 μm. All images were captured on the Echo Revolve R4 and images were cropped and resized using Affinity Designer. For each image, brightness and contrasts were uniformly adjusted relative to the brightest feature to balance exposure of each color channel.

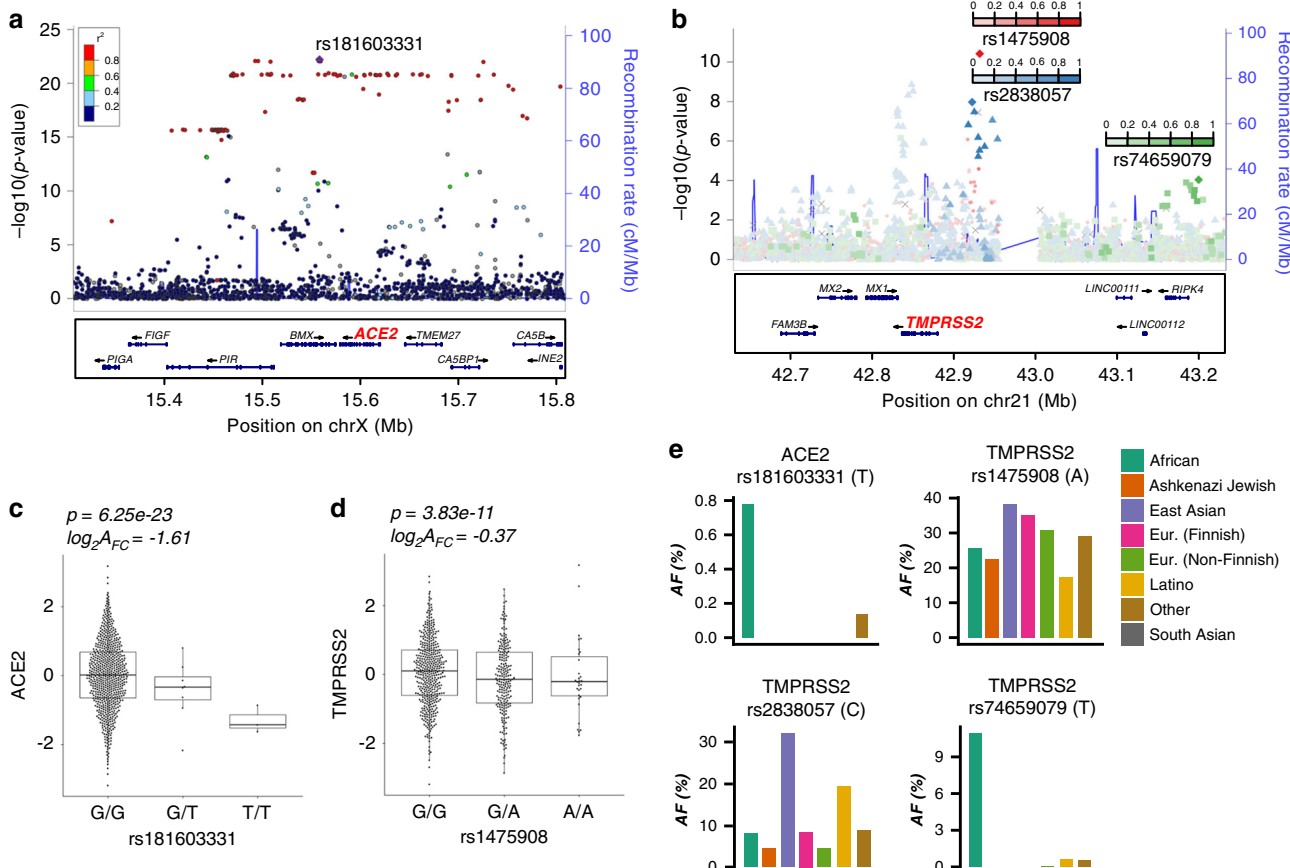

**Fig. 6 Nasal airway *ACE2* and *TMPRSS2* are regulated by eQTL variants. a** Locuszoom plot of *ACE2* eQTL signals. The lead eQTL variant (rs181603331) is highlighted with a purple dot. The strength of linkage disequilibrium (LD) between rs181603331 and other variants is discretely divided into five quantiles and mapped into five colors (dark blue, sky blue, green, orange, and red) sequentially from low LD to high LD. **b** Locuszoom plot of *TMPRSS2* eQTL signals. The three independent eQTL variants (rs1475908, rs2838057, rs74659079) and their LD with other variants ($r^2$) are represented by red, blue, and green color gradient, respectively. **c** Box plots of normalized *ACE2* expression among the three genotypes of the lead *ACE2* eQTL variant (rs181603331). log$_2$A$_{FC}$ = log$_2$ of the allelic fold change associated with the variant. (n: GG = 654, GT = 8, TT = 3). eQTL p-values were obtained from testing the additive genotype effect on gene expression using the linear regression model implemented in FastQTL. Benjamini–Hochberg correction was used to control for false discovery rate. **d** Box plots of normalized *TMPRSS2* expression among the three genotypes of the lead *TMPRSS2* eQTL variant (rs1475908). log$_2$A$_{FC}$ = log$_2$ of the allelic fold change associated with the variant. (n: GG = 432, GA = 218, AA = 31). eQTL p-values were obtained from testing the additive genotype effect on gene expression using the linear regression model implemented in FastQTL. Benjamini–Hochberg correction was used to control for false discovery rate. **e** Bar plots depicting allele frequencies of the *ACE2* eQTL variant rs181603331 and *TMPRSS2* eQTL variants (rs1475908, rs2838057, rs74659079) across world populations. Allele frequency data were obtained from gnomAD v2.1.1. Box centers give the median, upper and lower box bounds correspond to first and third quartiles, and the upper/lower whiskers extend from the upper/lower bounds up to/down from the largest/smallest value, no further than 1.5× IQR from the upper/lower bound (where IQR is the inter-quartile range). Data beyond the end of whiskers are plotted individually.

**Table 1 Results for multivariate models of *ACE2* and *TMPRSS2* expression.**

| Variable | Reference group | Partial $R^2$ (%) | Coefficient | SE | *t*-Statistic | *p*-value |
|---|---|---|---|---|---|---|
| *ACE2* | | | | | | |
| Age | n/a | 1.03 | −0.032 | 0.009 | −3.64 | 3.00E−04 |
| Interferon status | Low | 17.09 | 1.301 | 0.088 | 14.78 | 6.50E−43 |
| T2 inflammation | Low | 24.44 | −1.001 | 0.057 | −17.68 | 1.58E−57 |
| Sex | Male | 0.14 | 0.075 | 0.056 | 1.33 | 0.19 |
| Asthma | Healthy | 0.58 | −0.160 | 0.059 | −2.73 | 0.0064 |
| rs181603331 (G>T) | G/G | 1.20 | −0.635 | 0.162 | −3.92 | 9.75E−05 |
| | | | | | | |
| *TMPRSS2* | | | | | | |
| Age | n/a | 0.07 | −0.008 | 0.010 | −0.88 | 0.38 |
| Interferon status | Low | 0.07 | 0.087 | 0.098 | 0.88 | 0.38 |
| T2 inflammation | Low | 33.24 | 1.177 | 0.063 | 18.77 | 1.74E−63 |
| Sex | Male | 0.02 | −0.031 | 0.062 | −0.50 | 0.62 |
| Asthma | Healthy | <0.01 | 0.014 | 0.065 | 0.22 | 0.83 |
| rs1475908 (G>A) | G/G | 0.22 | −0.082 | 0.054 | −1.51 | 0.13 |
| rs74659079 (C>T) | C/C | 0.39 | 0.216 | 0.107 | 2.03 | 0.043 |
| rs2838057 (A>C) | A/A | 0.42 | 0.139 | 0.066 | 2.12 | 0.035 |

low frequency of this variant in our population. Increasing age and asthma diagnosis were both associated with small decreases in *ACE2* expression, although both variables accounted for <2% of the variance, and sex was not a significant predictor (Table 1).

Similar modeling of *TMPRSS2* expression found that T2-high status markedly increased expression, with an effect size 5.4 times larger than any other variable, capturing 33% of the total variation in *TMPRSS2* (Table 1). While statistically significant, the two *TMPRSS2* eQTL variants associated with increased expression exhibited small effect sizes totaling <1% of variance explained. All other predictors were not significant. Collectively, these modeling results confirm that both T2 and interferon inflammation are strong and antagonistic regulators of *ACE2* expression and show that T2 inflammation is the lone dominant driver of airway expression of *TMPRSS2*.

**Host response to CoV is similar to other respiratory viruses.** Our metagenomic analysis of RNA-seq data from the nasal brushings identified 113 subjects infected with a range of viral species, including 17 children infected with one of four different common coronavirus (CoV) species (OC43, JKU1, 229E, NL63) (Supplementary Data 5 and 6). This presented us with the opportunity to determine the host airway response to collective common CoV species and to contrast this response to that of other non-CoV respiratory viruses. We also reasoned this analysis would allow us to generate a reference host nasal airway response to common, non-severe CoV infections, which could be compared to host nasal airway responses to SARS-CoV-2 infections, likely to be generated by researchers soon, allowing investigators to discriminate between host responses that are pertinent or not to generating severe respiratory illness. To increase the likelihood that these subjects were experiencing an active viral infection, we limited our analysis to the 9 most highly CoV-infected subjects, comparing them to all subjects not infected with a virus ($n =$ 582). To allow us to investigate whether CoV infections stimulate a unique host response relative to other respiratory viruses, we established a viral infected group composed of 22 subjects highly infected with one of several viral species known to cause significant respiratory illness[27–33], including lower airway effects and asthma exacerbations, collectively referred to hereafter as the Other Respiratory Virus group (ORV). The ORV group included samples infected with human rhinovirus species C (HRV-C),

Influenza A, Influenza B, Orthopneumovirus, Metapneumovirus, Enterovirus, or Parainfluenza (Supplementary Data 6).

We first broadly examined the host viral response in the two virus-infected groups, evaluating the mean expression difference of the interferon response (tan) and cytotoxic immune response (purple) networks (discussed earlier; see Fig. 4a, b). We found that both networks were strongly upregulated among both the CoV and ORV virus-infected groups and that there was no significant difference in expression of these networks between the CoV and ORV groups (Fig. 7a, b).

To comprehensively explore host viral responses, we next performed a transcriptome-wide screen for differentially expressed genes (DEGs) in CoV- or ORV-infected groups compared to uninfected individuals (Supplementary Data 7). The two virus groups showed highly similar responses, with 94% of the differentially expressed genes with CoV infection also being identified as significant with ORV infection (false discovery rate [FDR] < 0.05 and $|\log_2 FC| > 0.5$). Highlighting the similarity of transcriptomic responses to viral infection in these two groups, the $\log_2 FC$ values for genes identified as significant in either group had a Pearson correlation of 0.95 ($p < 2.2e{-}16$; Fig. 7c). To further characterize whether the differences observed between the two groups were significant or a result of differences in statistical power from unequal sample sizes, we used Cochran's test for heterogeneity and discovered no significant difference for any of the DEGs that were identified in only the CoV- or ORV-infected groups after adjusting for multiple testing.

Upstream regulator analysis with Ingenuity Pathway Analysis (IPA) carried out separately on CoV and ORV infection response genes showed that the top cytokines and transcription factors that may regulate these infections are largely shared between the two virus families, which included IFNG, IFNA2, STAT1, IFNL1, and IRF7 (Fig. 7d). One inferred upstream regulator of CoV response genes, IL-6, which was also among the genes upregulated with viral infection (CoV $\log_2 FC = 2.0$, ORV $\log_2 FC = 3.0$, Fig. 7e), is especially noteworthy considering that an IL-6 blocking antibody therapy is currently under investigation for use in the treatment of COVID-19 illnesses[34]. In addition, we found *ACE2* among the shared upregulated genes, reinforcing its upregulation in the course of different respiratory virus infections ($\log_2 FC$ in CoV = 0.8, $\log_2 FC$ in ORV = 0.5, Fig. 7f). Together, these results suggest that host airway responses to common CoV infections are transcriptomically similar to the host airway responses to other respiratory viruses.

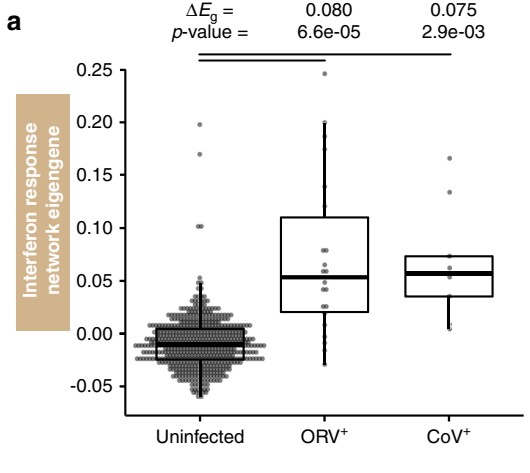

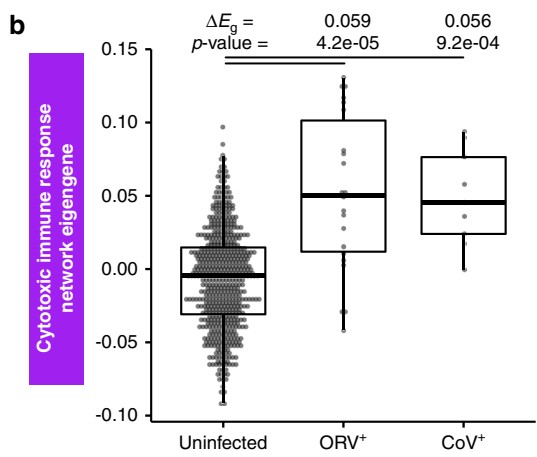

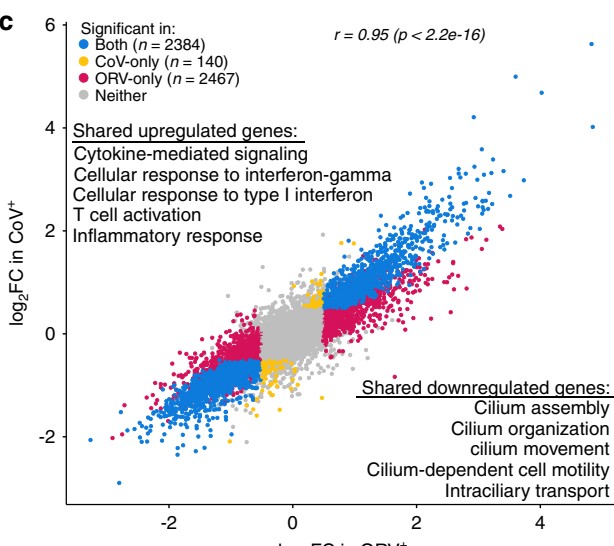

**d**

| Upstream Regulator | CoV | | ORV | |
|---|---|---|---|---|
| | Activation Z-score | Enrichment p-value | Activation Z-score | Enrichment p-value |
| IFNG | 12.24 | 1.24e–115 | 13.44 | 1.57e–136 |
| IFNA2 | 9.11 | 2.69e–71 | 8.94 | 1.93e–62 |
| STAT1 | 8.12 | 2.62e–69 | 8.57 | 6.46e–67 |
| IFNL1 | 7.79 | 1.12e–66 | 7.85 | 3.11e–49 |
| IRF7 | 8.37 | 3.85e–60 | 8.71 | 1.42e–44 |
| TNF | 10.46 | 1.65e–56 | 14.74 | 1.83e–122 |
| CD40LG | 5.81 | 1.03e–50 | 7.66 | 9.15e–68 |
| STAT3 | 5.05 | 4.79e–49 | 8.02 | 6.08e–69 |
| IRF1 | 6.81 | 9.67e–49 | 6.87 | 1.80e–44 |
| IL6 | 6.01 | 1.32e–29 | 7.89 | 5.62e–50 |

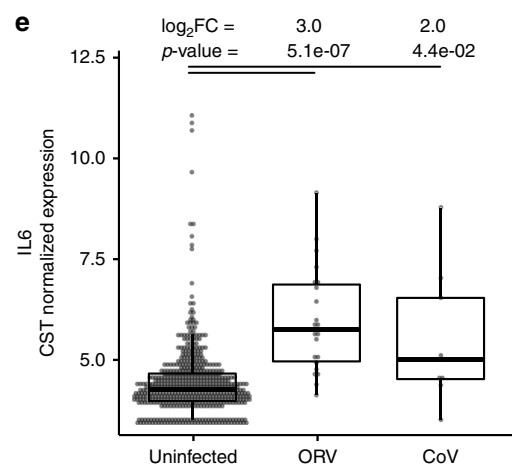

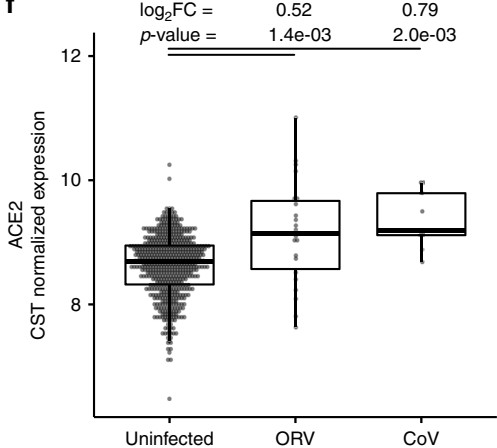

## Discussion

Although the high variability in clinical outcomes of COVID-19 illness is now well documented and multiple demographic and clinical traits have been associated with severe disease, little is known about the host biologic factors underlying this variability. In the current study, we reasoned that population variation in upper airway expression of the ACE2 receptor for SARS-CoV-2 and the virus-activating TMPRSS2 protease would drive infection susceptibility and disease severity. We, therefore, deployed network and eQTL analysis of nasal airway epithelial transcriptome data from a large cohort of healthy and asthmatic children to determine mechanisms associated with airway expression of these genes, and their relative power in explaining variation in the expression of these genes among children. We observed only weak associations with asthma status, age, and gender among children aged 8–21 years. Moreover, although we found that genetics does influence the expression of these genes, the effect of this variation was small in comparison to the marked influence of T2 cytokine-driven inflammation on both *ACE2* (downregulation) and *TMPRSS2* (upregulation) expression levels.

**Fig. 7 Host response to CoV strains are similar to other respiratory viruses. a** Box plots revealing a strong and equivalent upregulation of summary (eigengene [$E_g$]) expression for the interferon response network among ORV- and CoV-infected GALA II subjects ($n = 22$, 9 respectively), compared to uninfected subjects ($n = 582$). Differential expression analysis between infected and uninfected groups was performed with a two-sided $t$-test. **b** Box plots revealing a strong and equivalent upregulation of summary (eigengene [$E_g$]) expression for the cytotoxic immune response network among ORV- and CoV-infected GALA II subjects ($n = 22$, 9 respectively), compared to uninfected subjects ($n = 582$). Differential expression analysis between infected and uninfected groups was performed with a two-sided $t$-test. **c** Scatterplot showing the similarity in $\log_2$FC differential expression of genes in ORV ($x$ axis) and CoV ($y$ axis) infected individuals relative to uninfected subjects. The color of the points corresponds to the group in which each gene was identified as significant (FDR < 0.05, absolute $\log_2$FC > 0.5). The Pearson correlation coefficient of the significant genes (excluding the "Neither" category) is 0.95. Top enrichment terms for genes that were significantly differentially expressed in both virus infections (blue) are shown. **d** Top upstream regulators predicted by Ingenuity Pathway Analysis to be regulating the genes that were upregulated in CoV. Enrichment values for these CoV regulators, using the ORV upregulated genes are also shown. Enrichment $p$-values were obtained from IPA. No multiple comparison adjustment was performed. **e** Box plots revealing upregulation of *IL6* expression in virus-infected individuals ($n$: CoV$^+$ = 9, ORV$^+$ = 22, non-infected = 582). Differential expression analysis was performed with limma. Benjamini–Hochberg correction was used to control for false discovery rate. **f** Box plots revealing upregulation of *ACE2* expression in virus-infected individuals ($n$: CoV$^+$ = 9, ORV$^+$ = 22, non-infected=582). Differential expression analysis was performed with limma. Benjamini–Hochberg correction was used to control for false discovery rate. Box centers give the median, upper and lower box bounds correspond to first and third quartiles and the upper/lower whiskers extend from the upper/lower bounds up to/down from the largest/smallest value, no further than 1.5 × IQR from the upper/lower bound (where IQR is the inter-quartile range). Data beyond the end of whiskers are plotted individually.

We found an equally important role for viral-driven interferon inflammation in regulating levels of *ACE2* in the airway. In addition, through the study of in vivo upper airway CoV family member infections, we identified host transcriptional responses to CoV infections and inflammatory regulators of these responses. We were also able to confirm the strong up and downregulation of ACE2 protein levels by virus infection and IL-13, respectively. In total, our work provides a set of biomarkers that can be easily examined in COVID-19 patients, through analysis of nasal swabs, to determine the relative importance of these mechanisms and genes in governing susceptibility to infection, severe illness, and death.

Our scRNA-seq analysis of an in vivo nasal brushing from an adult asthmatic found *ACE2* expression, albeit at low frequency, primarily among basal, ciliated, and less mature, early secretory cells. A much higher portion of cells, representing all epithelial cell types, expressed *TMPRSS2*, although the low frequency of *ACE2*$^+$ cells resulted in very few dual *ACE2*/*TMPRSS2* expressing cells. However, we caution that a cell may not need to be *TMPRSS2*$^+$ to be susceptible to infection, since it has been demonstrated the TMPRSS2 protein is secreted from nasal airway epithelial cells[35]. We also caution that scRNA-seq data are known to exhibit biases in gene detection, and thus the level and frequency of *ACE2* expression across cells may be higher than we observe here. In line with this possibility, we observe more moderate levels of *ACE2* expression in our bulk RNA-seq data on nasal brushings. Although our results are based on a single adult donor, they correspond strongly to and validate the results of several recently published, large scRNA-seq surveys[36,37]. We also note that a recent report shows that airway epithelial expression of *ACE2* and *TMPRSS2* is similar between adults and children[38]. Importantly, we were able to show that ACE2 protein is expressed at the apical surface of the tracheal airway epithelium, particularly colocalizing with ciliated cells, a pattern we also observed among in vitro nasal mucociliary airway epithelial cultures.

Airway inflammation caused by type 2 cytokine production from infiltrating immune cells has a prominent role in the control of cellular composition, expression, and thus biology of the airway epithelium[12,14,24,39]. Moreover, while T2 airway inflammation is an important driver of T2-high asthma and COPD disease endotypes, it is also associated with atopy in the absence of lung disease, a very common phenotype in both children and adults. In fact, among the children in this study, we find that 43% of non-asthmatics were scored as T2-high based on expression profile, further substantiating the high prevalence of T2 airway inflammation outside of those with lung disease. Our data suggest that

airway epithelial *TMPRSS2* expression is highly upregulated by T2 inflammation, and specifically by IL-13. Both our network and single-cell data show that *TMPRSS2* is most prominent in less developed early secretory cells as well as in more mature mucus secretory cells. Based on our in vitro data, IL-13 upregulates *TMPRSS2* across nearly all types of epithelial cells, but the core of this effect appears to be in the metaplastic mucus secretory cells that are generated as a consequence of IL-13 signaling[15,16]. In fact, our network data suggest that, although *TMPRSS2* expression is highly correlated with that of a co-expressed network of mucus secretory genes characterizing normal, non-metaplastic, mucus secretory cells, it is correlated even more strongly with a network that characterizes mucus secretory cells undergoing IL-13-induced metaplasia. In contrast to enhanced levels of *TMPRSS2*, T2 inflammation, whether observed in vivo or induced with IL-13 stimulation, precipitated a marked reduction in levels of epithelial *ACE2*, thus making it difficult to predict how T2 inflammation might affect overall risk for a poor COVID-19 outcome. Germane to this question, a recent study of 85 fatal COVID-19 subjects found that 81.2% of these subjects exhibited very low levels of blood eosinophils[4]. Blood eosinophil levels are a strong, well-known predictor of airway T2 inflammation and were strongly correlated with T2 status in our study as well[12,24]. The strong downregulation of ACE2 protein at the apical surface of the epithelium by IL-13 is supportive that T2 inflammation effects on *ACE2* expression may be the dominant factor of any potential effect of T2 inflammation on outcomes. However, both in vitro experiments examining IL-13 effects on SARS-CoV-2 infection and empirical data on COVID-19 outcomes among T2-high and T2-low patients will be needed to determine whether this common airway inflammatory endotype ultimately protects against, exacerbates, or has no effect on COVID-19 illness. Given the higher frequency of T2 inflammation among asthmatic subjects, this population should be monitored especially closely, considering the enhanced risk of complications due to respiratory virus infection in those with asthma. We note that measurement of blood eosinophil levels could be used as an informative and more accessible (albeit less powerful) proxy for investigating the association between airway T2 inflammation and outcomes of COVID-19.

In addition to the strong negative influence of T2 inflammation on *ACE2* expression in the airway, we found an equally strong positive influence of respiratory virus infections on *ACE2* levels. Network analysis placed *ACE2* within an interferon viral response network suggesting that these cytokines are a driving force behind *ACE2* upregulation. This notion is supported by a recent

publication reporting *ACE2* as an interferon-stimulated gene in airway epithelial cells[37]. This information is interesting in several regards. First, it suggests that SARS-CoV-2 and other coronaviruses using ACE2 as a receptor could leverage the host antiviral response to increase the infectability of airway cells. Secondly, as data here and elsewhere show, asymptomatic carriage of respiratory viruses is common, especially in young children[20,40–43]. Children in the GALA II cohort included in this study ranged in age from 8 to 21 years; among them, we found 18% who were carrying respiratory viruses without illness. However, as we show in this and our previous study[20], even asymptomatic carriage of respiratory viruses exacts a fundamental change in airway epithelial expression and immune cell presence, including upregulation of *ACE2* expression. In fact, we find using in vitro mucociliary experiments that ACE2 protein is highly upregulated at the airway epithelial apical surface by HRV infections, suggesting co-infection with other respiratory viruses could prime a subject for SARS-CoV-2 infection. In determining outcomes, this potential detrimental influence of virus carriage may also be weighed against a potentially beneficial influence of virus carriage through a more potent cross serologic-immune defense in these individuals, especially if the virus carried is a coronavirus family member. Ultimately, the effect of current or recent virus carriage on COVID-19 outcomes will need to be determined by in vivo studies in patients, followed up with controlled in vitro studies of virally infected cells. Supporting a possible role for interferon responses in COVID-19 outcomes, a recent single-cell RNA-seq study found increased type I interferon response was a defining feature of patients with a severe vs. mild COVID-19 illness[44].

Our evaluation of genetic influences on airway *ACE2* and *TMPRSS2* expression revealed a single eQTL for *ACE2* and several common eQTL variants for *TMPRSS2*. While both the effect size and explanatory power of these variants paled in comparison to the influence of T2 inflammation and interferon signaling in multi-variable modeling of expression for these genes, the effect of these variants may still be significant enough to alter infection rates and/or illness severity, especially in the populations where these variants are most frequent. However, this assumes these variants will similarly function as eQTL variants in other populations, which will need to be confirmed. Relatedly, we found the allele frequency of these variants varies significantly across world populations. Therefore, we postulate that if *TMPRSS2* levels influence susceptibility to SARS-CoV-2, then infection risk may vary significantly across world populations as a result of the prevalence of these eQTL variants. Thus, future genetic studies of COVID-19 illnesses should pay particular attention to these eQTL variants.

A particularly vexing question regards the mechanisms that underlie the unusual severity of illness associated with SARS-CoV-2, especially when compared to most circulating respiratory viruses. Clearly, severe disease often entails the development of pneumonia, possibly resulting from an expanded tropism of SARS-CoV-2 to include lower lung airway and alveolar cells. The most severe patients also appear to experience an exuberant immune response, characterized as a cytokine storm[34], occurring with and possibly driving the development of acute respiratory distress syndrome (ARDS). This is in stark contrast to the mild cold symptoms triggered by other CoV species of the same family. To characterize the host nasal airway transcriptomic response to asymptomatic infection with common CoV family members, we leveraged the CoV-infected GALA II samples used in this study. We found that, despite a lack of symptoms, subjects exhibited a robust transcriptomic response, which included both epithelial innate immune pathways and activation of multiple immune cell types. Our analysis defined all genes differentially expressed by

these CoV infections and likely transcriptional regulators of these responses. Notably, we found that IL-6 was predicted to regulate airway responses to CoV and was itself upregulated in the airway with these infections. These data provide additional support for the ongoing investigation of tocilizumab (IL-6R blocking antibody) for the treatment of COVID-19 illnesses[34]. This robust CoV host response did not differ from the host response triggered by other respiratory viruses. Given the number of genes and pathways activated in these common, asymptomatic CoV illnesses, we believe this data will be valuable in deciphering the parts of the airway response to COVID-19 illnesses that are generic to common CoV illnesses, and those that underlie the severity of some COVID-19 illnesses.

To summarize, our data suggests that the strongest determinants of airway *ACE2* and *TMPRSS2* expression in children are T2 inflammation and viral-induced interferon inflammation, with limited but noteworthy influence from genetic variation. We caution that extrapolation of our results to other groups (e.g., adults and other racial groups) will require similar studies in these populations. Moreover, whether these factors drive better or worse clinical outcomes remains to be determined, but closely watching individuals with these airway endotypes in the clinical management of COVID-19 illnesses would be prudent.

## Methods

**Materials and correspondence**. Further information and requests for resources and reagents should be directed to and will be fulfilled by Max A. Seibold, Ph.D. (seiboldm@njhealth.org).

**Human subject information**. Under the Institutional Review Board (IRB) approved Asthma Characterization Protocol (ACP) at National Jewish Health (HS-3101), we consented a 56-year-old asthmatic subject, from which we collected nasal airway epithelial cells. The nasal airway cells were brushed from the inferior turbinate using a cytology brush and used for the scRNA-seq experiment described in Fig. 1. Nasal airway epithelial cells used for bulk RNA-seq network and eQTL analysis came from GALA II study subjects described below (*n* = 695, 681, respectively). Nasal airway epithelial cell ALI culture gene expression experiments used cells derived from GALA II study subjects (*n* = 5). Nasal airway epithelial cells used for ACE2 protein staining in IL-13/HRV experiments were from a single GALA II asthmatic donor. Human tracheal airway epithelial cells used for in vitro IL13 stimulation and scRNA-seq gene expression experiments were isolated from a single de-identified lung donor. Tracheal tissues used for the ACE2 immunofluorescence studies were obtained from two de-identified lung donors, from one of whom nasal airway epithelial cells were also obtained for ACE2 protein staining in IL-13/DAPT differentiation experiments. All lung donor sample was received from the International Institute for the Advancement of Medicine (Edison, NJ), and Donor Alliance of Colorado. The National Jewish Health Institutional Review Board approved our research on all lung donor tracheal/nasal airway epithelial cells and tracheal tissues under IRB protocol HS-3209. These cells and tissues were processed and given to us through the National Jewish Health (NJH) live-cell core, which is an institutional review board-approved study (HS-2240) for the collection of tissues from consented patients for researchers at NJH.

**GALA II study subjects**. The Genes-Environment & Admixture in Latino Americans study (GALA II) is an ongoing case–control study of asthma in Latino children and adolescents. GALA II was approved by local institutional review boards (University of California, San Francisco [UCSF], IRB number 10-00889, Reference number 153543, NJH HS-2627) and all subjects and legal guardians provided written informed assent and written informed consent, respectively[45,46]. A full description of the study design and recruitment has been described elsewhere[45–47], and we provide a summary here. A demographic table of this cohort is provided in Supplementary Data 8. Briefly, the study includes subjects with asthma and healthy controls of Latino descent between the ages of 8 and 21, recruited from the community centers and clinics in the mainland U.S. and Puerto Rico (2006–present). Asthma case status was physician-diagnosed, with additional requirements for two or more symptoms of coughing, wheezing, or shortness of breath within the 2 years prior to enrollment in the study. Subjects were not eligible if they were in the third trimester of pregnancy, were current smokers at the time of enrollment, or had at least a 10 pack-year smoking history. Recruited subjects completed in-person questionnaires detailing medical, environmental, and demographic information. Additional physical measurements were obtained, and subjects provided a blood sample for DNA extraction and later whole-genome sequencing (WGS). GALA II subjects that were part of this analysis were all recruited from Puerto Rico (*n* = 695). A nasal airway inferior turbinate brushing was used to collect airway epithelial cells from these subjects for

whole-transcriptome sequencing ($n = 695$). Network analyses were performed on all subjects with nasal brushing whole-transcriptome sequencing data ($n = 695$) and eQTL analysis was performed on the subset ($n = 681$) with whole-genome sequencing generated genotype data.

**Airway epithelial culture for gene expression analyses**. Primary human basal airway epithelial cells were expanded and differentiated at ALI in vitro following our established protocols[48], described here. Primary human basal airway epithelial cells were expanded on rat tail collagen-coated dishes in PneumaCult Expansion-Plus Medium (PEP) supplemented with Y-27632 (10 μM; PEP + Y) and harvested by trypsinization with FBS neutralization. The apical chambers of 6.5 mm transwell inserts were seeded at $4 \times 10^4$ cells/insert in PEP + Y media, and the basolateral chambers received 500 μL of PEP + Y media alone. Following 24 h incubation, Y-27632 was removed and the media changed to PEP media only. Once the epithelial basal cells reached confluence, the apical growth media was removed, and the basolateral medium was replaced with PneumaCult ALI medium, initiating day 0 of the ALI mucociliary culture system. Epithelial cultures were allowed to differentiate at ALI in vitro for 21–28 days under each respective stimulation described below. Paired tracheal ALI cultures for the IL-13 scRNA-seq analysis were samples of convenience. These cultures were used as mock gene-edited controls for a gene-editing experiment. Therefore, the basal cells from which these ALI cultures were derived were mock-edited with a CRISPR ribonucleoprotein (RNP) complex. Specifically, these basal cells were nucleofected with IDTDna's non-specific Alt-R® CRISPR-Cas9 Negative Control crRNA #1 complex (Catalog #: 1072544). The electroporated cell suspension was immediately transferred into a fresh culture vessel for recovery and expansion and differentiation as described above. ALI cultures derived from these basal cells were mock-treated or treated with 10 ng mL$^{-1}$ IL-13 in media (20 μL apical; 500 μL basolateral) for the final 10 days of differentiation (ALI days 11-21) before harvest and scRNA-seq analysis. In contrast, nasal ALI cultures used for the gene expression studies were from five randomly selected GALA II subjects were either stimulated with IL-13 for 72 h following completion of mucociliary differentiation (25 days) or were infected with human rhinovirus strain A16 for 4 h during the final 24 h of the 28 days of differentiation. Control cultures were only treated with media.

**Immunofluorescence staining of ACE2 protein**. To conduct the ACE2 immunofluorescence (IF) studies, we expanded and differentiated at ALI in vitro nasal basal airway epithelial cells from an asthmatic child in the GALA II study and a non-asthmatic, smoker child from the lung donor program and NJH cell core as described above. The asthmatic child ALI cultures were mock-treated, IL-13 treated (5 days), or HRV-A16 and -C infected (harvested 24 h post-infection) after >60 days at ALI. The non-asthmatic child ALI cultures were differentiated normally for 21 days, or differentiated for 21 days in the presence of IL-13 (daily treated), or DAPT (treated every other day) (Selleck Chemicals). ALI inserts from these donors were fixed in 4% PFA for 20 min at room temperature. Human trachea tissues from one child (same as the DAPT experiment donor) and one adult, non-smoker, a non-asthmatic donor (from the lung donor program described above) were fixed in 10% neutral buffered formalin for overnight. The tissues and ALI inserts were paraffin-embedded and sectioned onto glass microscope slides. Deparaffinization was performed with HistoChoice (Sigma-Aldrich), followed by a standard Ethanol dilution series (100, 90, 70, 50, and 30%), and antigen retrieval in 1× Antigen Unmasking Solution (100× Citric acid-based, VECTOR LAB) before blocking in Blocking Buffer (1× PBS, 3% BSA, and 5% FBS). Histology sections were incubated with primary antibodies KRT5 (1:2000; Biolegend), MUC5AC (1:500; ThermoFisher), ACT (1:2000; Sigma), and ACE2 (1:100; AF933; R&D Systems) for 1 h at room temperature followed by Alexa Fluorochrome-conjugated secondary antibodies (ThermoFisher) for 45 min at room temperature in the dark. For nuclear staining, cells were incubated with DAPI for 5 min at room temperature. All stained tissues were mounted with ProLong Diamond Mount Medium (Invitrogen) and imaged using Echo Revolve R4 fluorescence microscope. For quantification of ACE2 staining, in a blinded fashion, we took images of ×20 objective fields (12-13 fields per condition for the IL-13/HRV experiment and 5–6 fields per condition for the IL-13/DAPT differentiation experiments) and counted the number of apical ACE2-positive cells. All images were captured on the Echo Revolve R4 and images were cropped and resized using Affinity Designer (v1.8.3). For each image, brightness and contrasts were uniformly adjusted relative to the brightest feature to balance exposure of each color channel.

**Bulk RNA-sequencing of GALA II and culture Samples**. Total RNA was isolated from 695 GALA II subject nasal airway epithelial brushings using the AllPrep DNA/RNA Mini Kit (QIAGEN, Germantown, MD). Whole-transcriptome libraries were constructed using the KAPA Stranded mRNA-seq library kit (Roche Sequencing and Life Science, Kapa Biosystems, Wilmington, MA) from 250 ng of total input RNA with the Beckman Coulter Biomek FX$^P$ automation system (Beckman Coulter, Fullerton, CA) according to the manufacturer's protocol. Barcoded libraries were pooled and sequenced using 125 bp paired-end reads on the Illumina HiSeq 2500 system (Illumina, San Diego, CA). Bulk RNA-seq data for the nasal ALI cultures ($n = 5$) to measure *ACE2* and *TMPRSS2* levels reported in

Figs. 3b, c and 4g, h, were generated with KAPA Hyperprep Stranded mRNA-seq library kits (Roche Sequencing and Life Science, Kapa Biosystems, Wilmington, MA) and sequenced with a Novaseq 6000 using 150 bp paired-end reads.

**Whole-genome sequencing of GALA II Samples**. Genomic DNA was extracted from whole blood obtained from GALA II study subjects using the Wizard Genomic DNA Purification kits (Promega, Fitchburg, WI), and DNA was quantified by fluorescent assay. DNA samples were sequenced as part of the Trans-Omics for Precision Medicine (TOPMed) whole-genome sequencing (WGS) program[49]. WGS was performed at the New York Genome Center and the Northwest Genomics Center on a HiSeqX system (Illumina, San Diego, CA) using a paired-end read length of 150 base pairs, to a minimum of 30X mean genome coverage. Briefly, the sequences were mapped using BWA-MEM (v0.7.15)[50] to the hs38DH 1000 Genomes build 38 human genome reference with the options "-K 100000000 -Y". Variants were identified and called using the GotCloud[51] pipeline (v1.17). These variants were filtered for quality, and genotypes with at least 10× coverage were phased with Eagle 2.4[52]. Variant calls used in this study were obtained from TOPMed data freeze 8 variant call format files.

**Preparation of cultures for scRNA-seq**. Following stimulation experiments involving the tracheal airway epithelial ALI samples, apical culture chambers were washed once with phophate-buffered saline (PBS) and once with PBS supplemented with dithiothreitol (DTT; 10 mM), followed by two PBS washes to remove residual DTT. Cold active protease (CAP) solution (*Bacillus licheniformis* protease 2.5 μg mL$^{-1}$, DNase 125 U mL$^{-1}$, and 0.5 mM EDTA in DPBS w/o Ca$^{2+}$Mg$^{2+}$) was added to the apical culture chamber and incubated on ice for 10 min with mixing every 2.5 min. Dissociated cells in CAP solution were added to 500 μL cold FBS, brought up to 5 mL with cold PBS, and centrifuged at $225 \times g$ and 4 °C for 5 min. The cell pellet was resuspended in 1 mL cold PBS + DTT, centrifuged at $225 \times g$ and 4 °C for 5 min, and then washed twice with cold PBS. The final cell pellet was resuspended in PBS with 0.04% BSA for single-cell gene expression profiling with the 10× Genomics system. Sample capture, cDNA synthesis, and library preparation for 10 d IL-13 ALI stimulations were performed using protocols and reagents for 10× Genomics Chromium Single Cell 3′ v3 kit. Single-cell libraries were pooled for sequencing on an Illumina NovaSeq 6000.

**scRNA-seq of in vivo nasal brushing cells**. Nasal brush cells were dissociated from the brush using *Bacillus licheniformis* cold-active protease (10 mg mL$^{-1}$), EDTA (0.5 mM), and EGTA (0.5 mM) at 4 °C with vortex mixing, followed by enzyme neutralization with FBS. Red blood cell lysis was performed and cells were washed twice in 0.04% BSA/PBS. Cell concentration was adjusted to 400 cells μL$^{-1}$ for cell capture of ~8,000 cells using the 10× Genomics Chromium Next GEM Single Cell 3′ reagent kit chemistry. Sample capture, cDNA synthesis, and library preparation were performed following 10× Genomics Chromium Next GEM Single Cell 3′ v3 kit. The single-cell library was sequenced on an Illumina NovaSeq 6000.

**Preprocessing of GALA II RNA-seq data**. Raw sequencing reads were trimmed using skewer[53] (v0.2.2) with the following parameter settings: end-quality=15, mean-quality=25, min=30. Trimmed reads were then aligned to the human reference genome GRCh38 using GSNAP[54] (v20160501) with the following parameter settings: max-mismatches=0.05, indel-penalty=2, batch=3, expand-offsets=0, use-sarray=0, merge-distant-same-chr. Gene quantification was performed with htseq-count[55] (v0.9.1) using iGenomes GRCh38 gene transcript model. Variance stabilization transformation (VST) implemented in DESeq2[56] (v1.22.2) was then carried out on the raw gene count matrix to create a variance stabilized gene expression matrix suitable for downstream analyses. These RNA-seq data were analyzed and published for other analyses[57].

**Weighted gene co-expression network analysis**. To understand what biological mechanisms regulate the variation of nasal airway epithelial gene expression, Weighted Gene Co-expression Network Analysis[58] (WGCNA) v1.68 was performed on the VST matrix of 17,473 expressed genes. WGCNA analysis is a network-based approach that assumes a scale-free network topology. To adhere to the scale-free assumption of the constructed biological networks, a soft thresholding parameter (β) value of 9 was chosen based on WGCNA guidelines. Furthermore, minClusterSize was set to 20, deepSplit was set to 2, and pamStage was set to TRUE. A total of 54 co-expression networks were identified and described in Supplementary Data 2. WGCNA networks are referred to by different colors, and two of the identified networks, saddle brown, and tan were found by gene ontology enrichment with the tool enrichR to capture co-expressed genes that underlie type 2 (T2) inflammation and interferon inflammation, respectively. We hierarchically clustered all subjects based on the expression of only the genes in the saddle brown network and then used the first split in the dendrogram as the basis for assigning individuals to T2-high or T2-low categories (Supplementary Fig. 1a). Similarly, we hierarchically clustered subjects using only the genes in the tan network and then selected the dendrogram branches with the highest tan network expression as interferon-high and the other subjects as interferon-low (Supplementary Fig. 2a).

**Cis-eQTL analysis**. Cis-expression quantitative trait locus (eQTL) analysis was performed by following the general methodology of the Genotype-Tissue Expression (GTEx) project version 7 protocol[59], using the nasal RNA-seq data and WGS variant data from 681 GALA II subjects.

Namely, WGS variant data were filtered based on allele frequency (minor allele frequency [MAF] > 1%) and allele subject count (total number of subjects carrying minor allele ≥ 10). After filtering, 12,590,800 genetic variants were carried forward into the eQTL analysis. For expression data filtering and preparation, we first ran Kallisto[60] (v0.43.0) to generate transcript per million (TPM) values. We filtered out any genes that did not reach both TPM > 0.1 and raw counts > 6 for at least 20% of our samples. After filtering, 17,039 genes were then trimmed mean of M-values (TMM) normalized using edgeR[61] (v3.22.3). Finally, we applied an inverse normal transformation into the TMM-normalized expression values to render them suitable for eQTL analysis. To account for the global population structure, we ran ADMIXTURE[62] (v1.3.0) on the genotype data to create five admixture factors. We then ran Probabilistic Estimation of Expression Residuals[63] (PEER, v1.3) to create 60 PEER factors to utilize as covariates in the eQTL analysis along with admixture estimates, gender, age, body-mass index (BMI), and asthma diagnosis status. To perform cis-eQTL analysis, we utilized a modified version of FastQTL[64] that was provided by the GTEx project, with a MAF cutoff of 1% and a cis window of 1 Mb. The results from FastQTL's permutation pass were used to generate a genome-wide empirical p-value threshold for each gene, according to the GTEx method. In short, the empirical p-values were adjusted for multiple testing, and the p-value closes to the FDR threshold of 0.05 was used to compute a nominal threshold for each gene based on the modeled beta distribution from FastQTL. These gene-level significance thresholds ranged from 1e−6 to 1e−2. Furthermore, we performed a stepwise regression analysis to identify independent eQTL variants using QTLTools[65] (v1.1). Allelic fold change ($A_{FC}$) of the eQTL variant is computed using the aFC python script[66].

**Virus identification and quantification**. To identify individuals with asymptomatic virus infection at the time of sample collection, viral genomic sequences were recovered from bulk RNA-seq data using an in-house analysis pipeline. Raw RNA-Seq reads were quality and adapter trimmed with BBduk[67] (v38.79) at Q = 10. Read pairs mapping (concordantly or discordantly) to the human transcriptome or human genome were removed with Hisat2 (v2.1.0) and Bowtie2[68] (v2.4.1), respectively. Low entropy reads were removed with BBduk at entropy=0.6. The remaining reads were assembled with MetaSpades[69] (v3.14.0). Contigs shorter than 200 bp in length were removed and the filtered reads mapped to the contigs using Bowtie2. Per contig read counts were obtained with samtools[70] (v1.10). Contigs were matched to the human genome using BlastN[71] (v2.9.0) in MegaBlast mode with subject besthit enabled. Contigs yielding any results with an e-value better than $10^{-2}$ were removed. The remaining contigs were matched to the NCBI NT database (downloaded May 1st 2020) using BlastN with subject besthit enabled. An in-house tool was used to aggregate contigs using best-scoring combinations of High-scoring segment pairs (HSPs) from the Blast results, classify aggregated contigs, summarize read counts and filter results to include only sequences classified as respiratory viruses.

**Defining virus-infected groups and associated analysis**. To ensure we selected subjects that were experiencing an active host response to a viral infection, we required subjects from the GALA II cohort to have at least 1,000 viral reads in order to be included in the CoV vs. ORV analysis. This resulted in 9 CoV+ individuals. To generate a similar infection-control group, composed of subjects highly infected with different virus species, we pooled subjects with at least 1,000 reads of one of Enterovirus (n = 2), Influenza (n = 3), Metapneumovirus (n = 1), Orthopneumovirus (n = 3), Parainfluenza (n = 2), or Rhinovirus C (n = 11) for a total of 22 virally infected subjects in the Other Respiratory Viruses (ORV) category. All non-infected subjects (n = 582) based on the analysis described above, were used as a comparison group for the CoV+ and ORV+ groups.

In performing the CoV+ and ORV+ transcriptome-wide differential expression analyses, to account for the class imbalance of this experiment, log₂ count-normalized expression values in units of counts per million (calculated using edgeR v3.28.0) were passed to the arrayWeights function in the limma[72] R package (3.42.0). limma-voom was then used to perform differential expression analysis on the count-normalized expression values between the CoV+ and uninfected groups, as well as between the ORV+ and uninfected groups, controlling for age, gender, and asthma diagnosis status. Genes were required to have an FDR adjusted p-value < 0.05, and an absolute log₂FC > 0.5 to be considered significant. Based on these cutoffs, genes were classified as being shared if they were significant in both comparisons, or as CoV+-specific or ORV+-specific if significant in only one comparison. All significantly differentially expressed genes from either viral group were tested for heterogeneity of effect using Cochran's Q statistic as implemented by METASOFT (v2.0.0)[73]. Significance values were adjusted for multiple testing using the Benjamini–Hochberg[74] approach of the R function p.adjust.

**Gene set enrichment analysis**. To investigate enriched pathways within WGCNA networks (Fig. 2a) or within genes differentially expressed in CoV+ and/or ORV+ infected subject groups (Fig. 7c), we used Enrichr[75] to test for gene over-representation of network genes within a panel of annotated gene databases (Gene Ontology [GO] Biological Process [BP] 2018, GO Molecular Function [MF] 2018,

GO Cellular Component [CC] 2018, Ligand Perturbations from GEO up, Kyoto Encyclopedia of Genes and Genomes [KEGG] 2019 Human, and Reactome 2016). In addition, gene marker sets were obtained for each of 35 epithelial and immune cell types inferred from a recent scRNA-seq study on ~70,000 cells from human lung tissue samples obtained intraoperationally from three individuals[76]. The WGCNA networks reported in Fig. 2a were tested for overrepresentation within each of these marker sets, with an FDR cutoff of 0.05.

**Canonical pathway analysis**. We used QIAGEN's IPA program (v01–16; content version: 51963813, release 2020-03-11) to investigate canonical pathways and upstream regulators that were significantly enriched in one or both of the upregulated CoV+-specific or ORV+-specific gene sets.

**scRNA-seq analysis of the nasal epithelial brushing**. Initial processing of 10× scRNA-seq data, including cell demultiplexing, alignment to the human genome GRCh38, and unique molecular identifier (UMI)-based quantification was performed with Cell Ranger (version 3.0). Since the nasal brushing sample contains both epithelial and immune cell populations that have distinct expression profiles (e.g., Immune cell types express far fewer genes compared to epithelial cell types), clustering and cell-type identification were done in two stages: (1) an initial clustering with a less stringent filter to identify major epithelial and immune cell clusters was performed, (2) cells were reclustered with different independent filtering criteria for epithelial and immune cell types. All these analyses were performed using Seurat[77] R package (v3.5.1).

In the first stage, we removed cells with fewer than 100 genes detected or cells with greater than 25% mitochondrial reads. In addition, to remove possible doublets, we removed cells with higher than 6,000 genes detected and/or more than 20,000 UMIs. Lowly expressed genes (detected in fewer than four cells) were also removed. We then performed normalization using SCTransform[78] and ran principal component analysis (PCA) on the top 5,000 highly variable normalized genes. Clustering analysis was performed on the top 20 PCs using a shared nearest neighbor (SNN)-based SLM[79] algorithm with the following parameter settings: Resolution=0.8, algorithm=3. The single-cell expression profiles were visualized via embedding into two dimensions with UMAP[80] (Uniform Manifold Approximation and Projection), resulting in the identification of 11,157 epithelial cells and 229 immune cells based on known cell-type signatures.

In the second stage, we retained all the immune cells but removed epithelial cells with fewer than 1,000 detected genes. After this filtering, a combined 8,291 epithelial and immune cells were then normalized as in the first stage. Clustering analysis performed on the top 30 PCs with parameters (resolution=0.4, algorithm=1, k.param=10) identified 15 clusters. We then ran differential expression analysis using a Wilcoxon test implemented in Seurat's FindMarkers function on the CPM (count per million) normalized expression values to help with cell-type identification. Based on these cluster marker lists, two clusters were merged into a single secretory cluster, another two clusters were merged into a single ciliated cluster, and a final two clusters were combined as "indeterminate," based on the lack of defining marker genes for these clusters. Through this merging process, we arrived at 8 epithelial and 3 immune cell populations (Fig. 1a and Supplementary Data 1 and 9).

**Analysis of RNA-seq data from nasal cultures**. Raw sequencing reads were trimmed using a skewer with the following parameter settings: end-quality=15, mean-quality=25, min=30. Trimmed reads were then aligned to the human reference genome GRCh38 using HISAT2[81] (v2.1.0) using default parameter settings. Gene quantification was performed with htseq-count using the GRCh38 Ensembl v84 gene transcript model. After removing mitochondrial, ribosomal, and lowly expressed genes (those not expressed in at least two samples), we carried out differential expression analyses between paired IL-13-stimulated and control samples (n = 5 donors) and between paired HRV-infected and control samples (n = 5 donors) using the DESeq2 R package (v1.22.2).

**Analysis of scRNA-seq data from tracheal cultures**. As with the nasal brushing scRNA-seq data, 10× scRNA-seq data from ALI cultures grown from a single tracheal donor that were either mock- or IL-13 stimulated for 10 days were preprocessed using Cell Ranger (version 3.0, 10× Genomics). To safeguard against doublets, we removed all cells with the gene or UMI counts exceeding the 99th percentile. We also removed cells expressing fewer than 1,500 genes or for which >30% of genes were mitochondrial (genes beginning with MTAT, MT-, MTCO, MTCY, MTERF, MTND, MTRF, MTRN, MRPL, or MRPS), resulting in a total of 6,969 cells (2,715 IL-13-stimulated and 4,254 controls). After removing mitochondrial, ribosomal (RPL and RPS), and very lowly expressed genes (expressed in <0.1% of cells), we integrated expression data from IL-13 and control cells using the data set integration approach in Seurat[82] (Supplementary Fig. 5). For the integration analysis, we used the top 30 dimensions from a canonical correlation analysis (CCA) based on SCTransform normalized expression of the top 3000 most informative genes across the two datasets, where informativeness was defined by gene dispersion (i.e., the log of the ratio of expression variance to its mean) across cells, calculated after accounting for its relationship with a mean expression. We then carried out PCA on the integrated data set and used the top 20 components

for clustering and visualization. We used SNN (Louvain algorithm, Resolution=0.23, k.param=10) to cluster the integrated cells into 11 populations, which we visualized in two dimensions using UMAP (Fig. 3d). These clusters were assigned cell-type labels based their most upregulated genes, which were identified by carrying out differential expression analysis between each cluster and all others using Seurat's logistic regression (LR) test, in which cell treatment was included as a latent variable (Supplementary Data 10).

**Reporting summary**. Further information on research design is available in the Nature Research Reporting Summary linked to this article.

## Data availability

All raw and processed RNA-seq data used in this study are deposited in the National Center for Biotechnology Information/Gene Expression Omnibus (GEO) accession number GSE152004. Data originating from public repositories can be accessed at the following locations: gnomAD database (version 2.1.1): [https://gnomad.broadinstitute.org/]; GO Biological Process 2018 table [https://amp.pharm.mssm.edu/Enrichr/geneSetLibrary? mode=text&libraryName=GO_Biological_Process_2018]; GO Molecular Function 2018 table [https://amp.pharm.mssm.edu/Enrichr/geneSetLibrary?mode=text&libraryName= GO_Molecular_Function_2018]; GO Cellular Component 2018 table [https://amp.pharm. mssm.edu/Enrichr/geneSetLibrary?mode=text&libraryName=GO_Cellular_Component_ 2018]; Ligand Perturbations from GEO up table [https://amp.pharm.mssm.edu/Enrichr/ geneSetLibrary?mode=text&libraryName=Ligand_Perturbations_from_GEO_up]; Kyoto Encyclopedia of Genes and Genomes 2019 Human table [https://amp.pharm.mssm.edu/ Enrichr/geneSetLibrarymode=text&libraryName=KEGG_2019_Human]; Reactome 2016 table [https://amp.pharm.mssm.edu/Enrichr/geneSetLibrary?mode=text&libraryName= Reactome_2016]; Cell-type marker gene sets were obtained from Supplemental Table S4 in Travaglini, et al. [https://www.biorxiv.org/content/biorxiv/early/2020/03/20/742320/DC1/ embed/media-1.xlsx]; TOPMed freeze 8 variant calls are available from dbGaP accession phs000920.v2.p2; NCBI NT BLAST database was downloaded May 1st 2020 [https://ftp. ncbi.nlm.nih.gov/blast/db/v5/]; and NCBI Taxonomy was accessed May 25th [https://ftp. ncbi.nlm.nih.gov/pub/taxonomy/new_taxdump/new_taxdump.tar.gz].

## Code availability

The scripts used to perform the analyses described in the paper have been deposited to the GitHub repository: https://github.com/seiboldlab/ACE2_TMPRSS2_Airway.

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

## Acknowledgements

This work was supported by NIH grants (MAS) R01 HL135156, R01 MD010443, R01 HL128439, P01 HL132821, P01 HL107202, R01 HL117004, and DOD Grant W81WH-16-2-0018. The Genes-Environments and Admixture in Latino Americans (GALA II) Study and E.G.B. were supported by the Sandler Family Foundation, the American Asthma Foundation, the RWJF Amos Medical Faculty Development Program, the Harry Wm. and Diana V. Hind Distinguished Professor in Pharmaceutical Sciences II, the National Heart, Lung, and Blood Institute (NHLBI) [R01HL117004, R01HL128439, R01HL135156, X01HL134589]; the National Institute of Environmental Health Sciences [R01ES015794]; the National Institute on Minority Health and Health Disparities (NIMHD) [P60MD006902, R01MD010443, the National Human Genome Research Institute [U01HG009080] and the Tobacco-Related Disease Research Program [24RT-0025, 27IR-0030]. M.J.W. was supported by the NHLBI [K01HL140218]. Burchard NIH Support: T32 GM007546, R01 128439, R01 HL141992, R01 HL141845. Whole-genome sequencing (WGS) for the Trans-Omics in Precision Medicine (TOPMed) program was supported by the National Heart, Lung and Blood Institute (NHLBI). WGS for "NHLBI TOPMed: Gene-Environment, Admixture and Latino Asthmatics Study" (phs000920) was performed at the New York Genome Center (3R01HL117004-02S3) and the University of Washington Northwest Genomics Center (HHSN268201600032I). Centralized read mapping and genotype calling, along with variant quality metrics and filtering were provided by the TOPMed Informatics Research Center (3R01HL-117626-02S1; contract HHSN268201800002I). Phenotype harmonization, data management, sample-identity QC, and general study coordination were provided by the TOPMed Data Coordinating Center (3R01HL-120393-02S1, U01HL-120393, contract HHSN268201800001I). We gratefully acknowledge the studies and participants who provided biological samples and data for TOPMed. WGS of part of GALA II was performed by New York Genome Center under The Centers for Trans-Omics Genomics of the Genome Sequencing Program (GSP) Grant (UM1 HG008901). The GSP Coordinating Center (U24 HG008956) contributed to cross-program scientific initiatives and provided logistical and general study coordination. GSP is funded by the National Human Genome Research Institute, the National Heart, Lung, and Blood Institute, and the National Eye Institute. The authors wish to acknowledge the following GALA II study collaborators: Shannon Thyne, UCSF; Harold J. Farber, Texas Children's Hospital; Denise Serebrisky, Jacobi Medical Center; Rajesh Kumar, Lurie Children's Hospital of Chicago; Emerita Brigino-Buenaventura, Kaiser Permanente; Michael A. LeNoir, Bay Area Pediatrics; Kelley Meade, UCSF Benioff Children's Hospital, Oakland; William Rodríguez-Cintrón, VA Hospital, Puerto Rico; Pedro C. Ávila, Northwestern University; Jose R. Rodríguez-Santana, Centro de Neumología Pediátrica; Luisa N. Borrell, City University of New York; Adam Davis, UCSF Benioff Children's Hospital, Oakland; and Saunak Sen, University of Tennessee. The authors acknowledge the families and patients for their participation and thank the numerous health care providers and community clinics for their support and participation in GALA II. In particular, the authors thank the recruiters who obtained the data: Duanny Alva, MD; Gaby Ayala-Rodríguez; Lisa Caine, RT; Elizabeth Castellanos; Jaime Colón; Denise DeJesus; Blanca López; Brenda López, MD; Louis Martos; Vivian Medina; Juana Olivo; Mario Peralta; Esther Pomares, MD; Jihan Quraishi; Johanna Rodríguez; Shahdad Saeedi; Dean Soto; and Ana Taveras. The content is solely the responsibility of the authors and does not necessarily represent the official views of the National Institutes of Health.

## Author contributions

Conceptualization, M.A.S., S.P.S., and P.D.; methodology, S.P.S., P.D., E.P., and N.D.J.; software, S.P.S., P.D., E.P., and N.D.J.; validation, Y.L., J.L.E., M.T.M., J.D.N., E.G.P., and M.A.S.; formal analysis, P.D., N.D.J., and S.P.S.; investigation, M.A,S, Y.L., J.L.E., M.T.M., J.D.N., C.L.R., and E.G.P.; resources, M.A.S., M.E.W., A.C.M., C.E., S.S., V.M., E.M.W., S.H., D.A.N., S.G., M.C.Z., G.A., H.M.K., K.M.R., R.K., S.O., and J.R-S.; writing—original draft, S.P.S., P.D., N.D.J., and M.A.S.; writing—review and editing, S.P.S., P.D., N.D.J., A.C.M., E.G.B., E.M.W., K.M.R., R.K., and M.A.S.; visualization, S.P.S., P.D., N.D.J., and M.A.S.; supervision, S.P.S., P.D., and M.A.S.; funding acquisition, E.G.B. and M.A.S.

## Competing interests

The authors declare no competing interests.
