## [Peer Review File · Nature Communications]

REVIEWER COMMENTS

Reviewer #1 (Remarks to the Author):

The manuscript uses a collection of banked nasal samples to look at the regulation of key pathways associated with SARS-CoV-2 infection, primarily focused on ACE2 and TMPRSS2. They find that ACE2 is upregulated by IFN signaling pathways, while TMPRSS2 is upregulated more by Type 2 signaling (which appears to downregulate ACE2). Some genetic analyses are performed, though the immune modulation appears to play a larger role than genetic variation. Finally they look at transcriptional responses to pooled non-SARS human coronavirus infections and identify a “cytotoxic signature” relative to HRV infections.

There are some useful data here, particularly regarding the SNPs influencing ACE and TMPRSS2, which, while not major factors in population variation are still helpful data for the literature given the interest in this area. Additionally the immune modulation (T2 vs. interferon) driving changes in receptor expression may play important roles physiologically. The genetic analysis based on such a large set of WGS is a major strength. However I have concerns about the interpretation (and overinterpretation) of some key data as detailed below.

1) The transcriptional comparison of “all coronaviruses” vs. rhinoviruses is of questionable significance. Rhinoviruses are known to be somewhat more restricted to the URT and to not induce robust adaptive memory responses. Given that they had 20 influenza-infected individuals, why wasn't this the more relevant comparison group?

2) Further, pooling all coronaviruses is of questionable utility. Of even more questionable utility is the idea that these might represent a proxy for SARS-CoV-2 biology—the sentences that suggest this should be softened and revised thoroughly. It might be of some interest to break out the data from NL63 specifically since it uses the same receptor as SARS, but even then it is still biologically distinct. Even if the initial analysis and statistics is pooled, breaking them out in the supplemental data would be helpful

3) Related to this, the abstract states that coronaviruses stimulate “a more pronounced cytotoxic immune response relative to other respiratory viruses”—this is not true, at best it is only relative to HRV. The language is similarly sloppy elsewhere and suggests a much larger conclusion than can be drawn (see also line 409).

4) The gene set enrichment analysis that underlies some of the “cytotoxic signature” presented in 6f and 6g do not look particularly convincing and no statistical significance is provided, between the shared and CoV specific signatures.

5) The introduction contains a sensational and overly earnest justification for the study. The statement “children are also thought to be highly susceptible to infection” is not cited and likely for good reason, as it is not supported by any reputable research. This is followed by the statement “Moreover, recent data suggest that 38% of COVID-19 cases occurring in children are of moderate severity and 5.8% are severe or critical”—this cites a single, poorly written and designed study which itself acknowledges that the 5.8% number comes from suspected cases rather than confirmed cases “There were more severe and critical cases in the suspected than confirmed category in this study. However, it remains to be determined if these severe and critical cases in the suspected group were caused by 2019-nCoV or other pathogens (eg, RSV). It may become clearer because the epidemic is quickly unfolding.” Further, their reported adult severity rate is significantly higher than rates reported in better designed and more reputable studies suggesting that this group's selection criteria did not allow for a reasonable estimate of the infection fatality rate. There is no need for cherry-picking poorly designed studies that are completely out of line with the bulk of reported data to try to “juice” the significance of the current study.

6) This sentence seems tautological to me: line 102 “as many as 70% of genes expressed by a tissue or organ are under genetic control.”

Reviewer #2 (Remarks to the Author):

This is a peer-review of paper by Sajuthi et al, which examined gene expression patterns of nasal epithelial cells obtained from brushes in over 500 children. They found that gene expression of ACE2, the putative receptor for SARS-CoV2, was significantly related to interferon response network and genes involved in the cytotoxic immune response; whereas TMPRSS2 was related to T2 mucus secretory network.

Major Comments:

1. One major limitation is that the investigators used nasal epithelia of children (who generally do not become sick with COVID19) rather than the more relevant adult tissues. Further, it would have been desirable to examine gene expression patterns in bronchial epithelia rather than nasal epithelia as the more severe COVID-19 cases involve lungs.
2. There are no specific experiments using SARS-CoV2 and as such the data derived from other respiratory viruses are questioned for their relevance to COVID-19. Viruses behave differently and elicit different immune responses.
3. There are no protein data to corroborate gene expression information. Protein data would have strengthened the observation.

Detail Comments for each section are denoted below:

Introduction

Page 6. Line 103-105. This seems that is not necessarily connected to SARS coronaviruses and host genetics. How is HRV similar to SARS-CoV-2? Are there any specific similarities? Both viruses have S proteins? or part of the genome is comparable?

Page 6. Line 105-107. The existence of genetic polymorphisms is only relevant in presence of phenotype variability. Given the phenotype variability of COVID-19 in children and the presence of genetic variations in the population (even within in cohorts of similar individuals), then it is possible that functional variants could explain differences in severity. I suggest reviewing this justification.

Page 6. Line 114-119. I suggest removing this.

Methods

General remarks: Overall methods are well described; however, the organization could be improved. A concise version of the methods in the main manuscript would be beneficial to the reader. Specific details can be included in the supplementary files.

The authors should correctly refer to the abbreviations, additionally, sample sizes use for each analysis should be clear from the beginning.

Page 25. Line 551 - 552. Please provide here the number of individuals used from the GALA cohort (line 551 page 25) and the number of subjects used use for ALI.

Page 26. Line 574. Which measures of spirometry were obtained used for this study? Were they used

in any way?

Page 26. Line 575. Provide abbreviation for Whole Genome Sequencing when it is first introduced.

Page 27. Line 583. Describe the meaning of the abbreviations (i.e.: ALI and WGS) when they are first introduced.

Page 27. Line 591. How many ALI samples were BULK RNAseq? All GALA cohort or 5 subjects? Please clarify and provide n from the beginning.

Page 28. Line 607 - 608. I suggest including a small summary of the genotype by sequence approach used.

Page 28. Line 612 - 613. ALI description protocol and culture should be provided earlier in the methods when the concept is first introduced.

Page 28. Line 616 - 617. Was there a criterion used to select these 5 subjects? If so, please mention it.

Page 31. Line 575 - 576. How did the authors determine which genes 'underlie' T2 and interferon inflammation? Were only these genes used for the hierarchically clustered of the subjects? Explain.

Page 32. Line 700 - 703. What was the p value threshold for the eQTL analysis? Considering the sample size and MAF, I wonder if you should assess the rare variant with a more stringent p value? I don't think this will change the bottom line of the manuscript. Also, did the authors set a LD window and MAF for the stepwise regression analysis? Please include this information.

Page 33. Line 721. Describe abbreviation of CoV

Page 33. Line 741. "The function arrayWeights function" needs to be corrected.

Page 34. Line 757 - 760. "For cell type enrichments within WGCNA networks reported in Figure 2a, we tested for overrepresentation of network genes within gene marker sets (FDR < 0.05) for each of 35 epithelial and immune cell types inferred using scRNA-seq of human lung tissue."

This sentence is too complicated and hard to follow, please re-write. In addition, the authors should provide a brief description of the scRNA-seq of human lung tissue used to infer the epithelial and immune cell types.

Page 34 - 35. Line 765 - 766. Are the authors referring to testing differential expression between immune cells and the rest of the cell types? How many cell types (other than immune) were in the data?

Page 37. Line 815 - 816. Is the title "Analysis of bulk RNA-seq data from IL-13 and HRV infected ALI nasal airway epithelial cultures", refers to "IL13-stimulated and HRV infected"?

Page 38. Line 857. CODE AVAILABILITY information is missing, please provide repository use to deposit the codes.

Results

General remarks: The authors aimed to fully described the results of this study. However, in multiple instances the sentences were too long and wordy, particularly at the beginning of this section. A study demographic table should be provided.

Page 9. Line 173 - 174. Are these the top expression programs represented by the networks?

Page 9. Line 176 - 185. Run-on sentences. I suggest providing one clear example instead of these three.

Page 10. Line 204-210. Although figure 3 looks convincing; It is a strong statement to say that this experiment can assess causality between T2 inflammation and TMPRSS2/ACE2 expression.

Page 12. Line 247. "Metagenomically" is not the correct way to express genome-wide meta-analysis. Please correct this.

Page 12. Line 260 - 261. Sure, but only HRV-A viral infections, the effect of other viruses remains to be tested. Please clarify that the results are specific to the virus been used in the experiment.

Page 13. Line 269 -270. What p threshold was used to identify associated variants?

Page 13 - 14. Line 283 - 298. This looks more like interpretation rather than results. I suggest removing these lines from the results.

Also, different AF in different population don't suggest that if TMPRSS2 levels influence susceptibility to SARS-CoV-2, then genetics may play a significant role in infection risk. Different population may have different AF of the variants simply because of genetic drift or population structure. Without more supporting information on other cohorts this is not a correct statement.

Instead, the results suggest that even within a cohort similar ancestry, expression vary due to eQTLs (at least partially), hence genetic determinants of TMPRSS2 expression may play a significant role in infection risk.

Page 15. Line 314 - 316. "While the ACE2 eQTL variant, rs181603331, was associated with a notable decrease in ACE2 levels, it only accounted for 1.2% of the variance, reflecting the low frequency of this variant in our population." How do the authors reach this conclusion without calculating the additive variant component?

It is more likely that the effect of the single SNP is inflated due to its low MAF. Unlike common variants, rare ones have a larger effect size on a trait. The heritability of the expression is not presented here; therefore, you don't know how much of the expression is due to overall genetics, or whether if 1.2% for a single SNP can be considered a small proportion of what can be capture from the heritability. Please review your interpretation and move it to the discussion.

Page 18. Line 386-387. I really like the comparison between CoV and HRV. However, the statement at the end "These results suggest that while CoV infections are highly similar to HRV infections, they likely elicit an enhanced cytotoxic immune response" seems a bit strong, I suggest been cautious when referring to the similarities between type of viruses, particularly in the result section.

Discussion

General remarks: Limitations of this study are not fully stated in the discussion. HRV infection experiments results and its implications with COVID-19 infection could be expanded.

Page 19, Line 396 - 405. I suggest reviewing and summarizing this sentence.

Page 19. Line 414 - 426. The main focus of this study are children, however the scRNAseq was obtained from a middle-aged adult. It is quite possible that expression changes with age. How do you think this difference could affect your conclusions.? You should mention this limitation. In addition,

authors should acknowledge that scRNAseq from one subject may not be representative of the population.

Page 20. Line 442 - 449. I suggest reviewing and summarizing this sentence. What is the meaning of "complicating expectations" in this context?

Page 21. Line 449 - 456. Are there other signs of T2 inflammation in severe or fatal COVID-19 cases that could further support your hypothesis?

Page 23. Line 491 - 489. Why did the authors stated that variants may have strong effects sizes in a larger population? This is not likely. Overall, as a study cohort gets larger, effect sizes will become smaller because of a wider range of phenotypes. Please review your statement.

eQTLs are very common across the genome, however only few eQTLs were identified on the GALA cohort, which suggest genetics have a small influence on the natural variation of expression. However, it is also possible that the eQTL analysis was underpower, therefore missing relevant eQTLs. This limitation needs to be acknowledged by the authors.

"More frequent variants" are called common variants. Please change.

Reviewer #3 (Remarks to the Author):

The manuscript describes expression of ACE2/TMPRSS2, two SARS-CoV-2 related genes, in human nasal epithelium and predicts potential mechanisms regulating their expression. The authors mine next-generation sequencing data (bulk and single cell RNA-seq, whole genome sequencing, cis-eQTL analysis, metagenomic analysis), to demonstrate the cellular expression pattern of ACE2/TMPRSS2 in nasal epithelial cells and immune sub-types. They identified a number of factors associated with ACE2 or TMPRSS2 expression in nasal epithelium, including type 2 inflammation, "interferon" response, eQTL variants, and respiratory virus infection. Using multivariate regression analysis, the authors estimate the relative importance of these factors to ACE2/TMPRSS2 expression. While of interest, relationships among QTL, ACE2/TMPR and patient endotypes are not identified in the present work. Novelty of the present work may be impacted by several recent publications, see (e.g., Sungnak et al., Nat Med 2020, Ziegler et al., Cell 2020, Jackson et al., JACI 2020, etc.) who identified a number of the findings presented in this paper, including the associations of type 2 inflammation, type 1 interferon, and asthma with ACE2/TMPRSS2 expression in nasal epithelium.

Specific Comments:

1. 10X 3' v3 scRNA-seq of nasal brushing from a single asthmatic subject is provided. Single cell expression of ACE2/TMPRSS2 in nasal cells from healthy donors and those with allergic inflammation were reported in several studies (e.g., Sungnak et al., Nat Med 2020; Ziegler et al., Cell 2020). These studies also showed cellular expression patterns of ACE2 in scRNA-seq data from in vitro nasal epithelial cell cultures. Insights added by the scRNA-seq from a single asthmatic subject in the present work are unclear.

2. The authors used scRNA-seq data to identify cell types. The authors should specify the markers they used to define cells. What defines and distinguishes different secretory cells? A two-stage analytic

approach was used to identify cell types. In the first stage, cells passed a QC filtering and clustering analysis identified epithelial or immune lineages of the cells. A second cell filtering was used to remove about 3,000 (~ 30%) epithelial cells. The reason for this second filtering was not well explained. It is not clear whether the expression of ACE2 or TMPRSS2 in those “removed” epithelial cells affected the overall patterns?

3. The authors used gene co-expression network and hierarchical clustering of bulk RNA-seq from nasal brushings from 695 healthy and asthma donors (children of Latino descent in Puerto Rico, 331 control and 364 donors with asthma), to identify in vivo relationships between airway type 2 inflammation and TMPRSS2/ACE2: samples with high T2 inflammation showed high TMPRSS2 but low ACE2 expression. TMPRSS2 was highly correlated with a gene set enriched for goblet cell markers and T2 cytokines IL13; ACE2 expression was negatively correlated with this gene set. Using in vitro stimulation of paired air-liquid interface (ALI) airway cultures, the authors showed that IL13 increased TMPRSS2 and decreased ACE2. Using scRNA-seq data from tracheal epithelial cultures that were chronically treated (10 days) with IL-13 or control media, the authors showed cell specific patterns of TMPRSS2 and ACE2 regulation by IL-13. The pathway and cell type enrichment analysis of the identified gene sets is important in this analysis. For each gene set, only one or two selected pathways or cell types were listed. Were the listed pathways/cell types the most enriched ones to best characterize each gene set?

4. The gene co-expression network identified ACE2 as highly correlated with that of a gene set enriched for interferon and viral responses. Hierarchical clustering analysis using this “interferon” gene set partitioned samples into “interferon-high” and “interferon-low” groups. ACE2 expression was increased in the “interferon-high” group vs. “interferon-low”. The “interferon-high” and “interferon-low” groups were defined on the expression of a de novo gene set which was predicted to be associated with interferon responses based on pathway enrichment analysis. The true interferon status of the subjects in these two groups merits an independent validation. A recent study demonstrated decreased nasal epithelial expression of ACE2 in children with asthma (Jackson et al., JACI 2020). The present analysis used data from both healthy and asthmatic subjects, thus, it is not clear what the distribution of asthmatic subjects was in the “interferon-high” and “interferon-low” groups, and the impact of asthma status on the findings was not entirely clear.

5. Using scRNA-seq data from in vitro epithelial cultures, the authors concluded that expression patterns of ACE2/TMPRSS2 were similar to those from “in vivo” nasal scRNA-seq data. However, in vivo scRNA-seq was represented by nasal brushing from a single 56-year-old asthmatic donor and the in vitro scRNA-seq used tracheal epithelial cells from a single de-identified lung donor (the lung condition and age information were unknown). Two different methods (from pre-processing to clustering to differential expression) were used to analyze these two scRNA-seq data. Thus, there are concerns regarding the cell type assignments in these distinct tissues.

6. Using cis-eQTL analysis of bulk RNA-seq of nasal airway epithelial brushings and whole-genome sequencing of 681 subjects, the authors identified eQTL variants associated with ACE2 or TMPRSS2 expression. The frequencies of those eQTL variants among eight different populations were investigated using the public gnomAD genetic variation database. The authors statement that “if TMPRSS2 levels influence susceptibility to SARS-CoV-2, genetics may play a significant role in infection risk and that this risk will vary significantly across world populations” is highly speculative. It is presently unclear whether ACE2/TMPRSS2 levels influence COVID infection or severity. The present eQTL variants were identified using data from both asthma and healthy donors, thus, it is unclear what the impact of asthma on expression is. Associations were identified using data from a single

population (children of Latino descent between the ages of 8 and 21 in Puerto Rico) and thus, merits validation in a separate cohort.

7. Using metagenomic analysis of the nasal epithelial brushings bulk RNA-seq data from healthy and asthmatic children, the authors identified subjects harboring reads for a respiratory virus. "To discriminate CoV-enhanced responses from those that are more general to respiratory viruses", the authors compared the transcriptomic profiles of subjects likely infected with a coronavirus with the profiles of subjects likely infected with human rhinovirus species, and those not infected with a virus. The metagenomic analysis also identified a comparable number of subjects likely infected with influenza. Inclusion the influenza group for comparison might improve identification of potential CoV enhanced or specific responses.

The authors state that "even asymptomatic carriage of respiratory viruses exacts a fundamental change in airway epithelial expression and immune cell presence, including upregulation of ACE2 expression". Nevertheless, in the present multivariate regression, this factor is not included. What is the relative importance of asymptomatic carriage of virus on ACE2 compared to other factors?

Minor comments:

(1) In line 373 of page 17, citation is missing.

(2) There are missing symbols in Methods, such as "Trans□Omics" on page 27

Reviewer #1

The manuscript uses a collection of banked nasal samples to look at the regulation of key pathways associated with SARS-CoV-2 infection, primarily focused on ACE2 and TMPRSS2. They find that ACE2 is upregulated by IFN signaling pathways, while TMPRSS2 is upregulated more by Type 2 signaling (which appears to downregulate ACE2). Some genetic analyses are performed, though the immune modulation appears to play a larger role than genetic variation. Finally they look at transcriptional responses to pooled non-SARS human coronavirus infections and identify a “cytotoxic signature” relative to HRV infections.

There are some useful data here, particularly regarding the SNPs influencing ACE and TMPRSS2, which, while not major factors in population variation are still helpful data for the literature given the interest in this area. Additionally the immune modulation (T2 vs. interferon) driving changes in receptor expression may play important roles physiologically. The genetic analysis based on such a large set of WGS is a major strength. However I have concerns about the interpretation (and overinterpretation) of some key data as detailed below.

Response: We appreciate the reviewer recognizing the usefulness of our data and analysis to the scientific community studying SARS-CoV-2. In retrospect we agree with the reviewer that some data in the manuscript was needlessly overinterpreted, and as you will see detailed below we have corrected this in our revised manuscript. We thank you for your thoughtful review.

1) The transcriptional comparison of “all coronaviruses” vs. rhinoviruses is of questionable significance. Rhinoviruses are known to be somewhat more restricted to the URT and to not induce robust adaptive memory responses. Given that they had 20 influenza-infected individuals, why wasn't this the more relevant comparison group?

Response: The rationale for, methods used, and interpretation of results in former section 6 of the paper entitled, “Coronavirus infections drive an enhanced cytotoxic immune response,” was questioned and critiqued in some form by all three reviewers. Therefore, we formulate one central response and explanation of revision for this paper section here. To be clear, this paper was designed to elucidate genetic and immune/biologic factors most important for controlling population variation in airway expression of the SARS-CoV-2 infection critical genes *ACE2* and *TMPRSS2*, as well as to better understand the airway epithelial biology related to these genes. As such the section 6 analysis was always ancillary to the main manuscript focus. Nonetheless, we viewed the opportunity to examine the human *in vivo* airway response to coronavirus infections and to contrast these responses to that of other viruses as a significant opportunity. The first premise for this analysis was that, based on their highly shared genome sequence and genetic structure, coronaviruses as a family would generate host responses that were similar and that some of these responses would differ from host responses generated by other respiratory virus species. The second premise, providing relevance to a study focused on SARS-CoV-2, was that some of these coronavirus family specific or enhanced host responses would also be shared with SARS-CoV-2 infections (surmised but not tested). We believe these assumptions are reasonable and given the limited data on coronavirus host airway effects, coupled with the ease of analyzing a sample of convenience, we reasoned this was a worthwhile task. Sample selection for this analysis is complicated, and both reviewer 1 and 3 suggested the use of a virus species other than HRV, namely influenza, as a more appropriate comparator. Our selection of HRV as the

comparator was based both on the fact that HRV was the most common infection in the cohort (allowing a robust comparison) and that HRV species infections have been associated with severe illnesses in young children, most notably triggering asthma exacerbations¹. However, at the reviewer's suggestion we now see the value in comparing to influenza infections, given their ability to trigger severe lower lung illnesses similar to SARS-CoV-2.

Unfortunately, we have recently discovered that only 5 of 20 originally called influenza infected samples were true infections. This is because a bug in the VirusFinder 2 software we employed allowed non-viral contigs to be misclassified as influenza. We caught this error as we have been developing our own analytical pipeline to identify viruses from RNA-seq data. Our new virus detection pipeline is complete and we have rerun all GALA samples. While the overwhelming majority of virus calls originally in the paper were consistent between the pipelines we did find a low number of conflicting calls which we have hand-curated to ensure accuracy. As noted above as significant portion of the small number of VirusFinder 2 miscalls were false-positive influenza calls. No important results in the paper were altered by these changes, but as we now only have 5 true influenza infected samples we are unable to run the influenza comparison requested by reviewers.

Instead to address the reviewer concerns over: (1) using a comparator virus species that can cause severe illnesses, and (2) in broadening the scope of viruses compared to the CoV family, we designed a new section 6 analysis for the paper. Specifically, we used a new severe, non-CoV virus comparator group that includes samples infected with human rhinovirus C (n=11), Influenza A (n=3), Orthopneumovirus (n=3), Metapneumovirus (n=1), Enterovirus (n=2), and Parainfluenza (n=2) (total n=22). References for these viruses causing severe illnesses are included in the paper. In this new analysis comparing genes differentially expressed with CoV family infected samples vs. uninfected samples, to genes differentially expressed with "other respiratory viruses" (ORV) infected samples vs. uninfected samples, we found a similarly upregulated cytotoxic immune response in both the CoV and ORV groups. This is in contrast to the more upregulated cytotoxic response for CoV infected samples we observed in the original manuscript version when the virus comparator group only included HRV infected samples. In fact, this revised analysis indicates the CoV and ORV host responses are strikingly similar, with no differences we immediately recognize that are of consequence. We believe these "negative" findings are important to report for the following reasons: (1) our analysis provides strong evidence against novel, strong host viral airway responses that are common to CoV family members and differ from other virus species, (2) our data is striking in that despite representing clinically asymptomatic viral infection, the vast majority of well-recognized host airway responses are triggered, suggesting that other mechanisms possibly outside of the airway (systemic?) are responsible for clinical symptoms and severity resulting from viral illnesses, (3) related to reason (2) we believe that our CoV asymptomatic illness dataset and host response analysis would be of high utility to compare and contrast to the likely many SARS-CoV-2/COVID-19 illness host airway response datasets which will come out of the scientific community. This comparison would allow these investigators to more quickly pinpoint responses that are unique to SARS-CoV-2/COVID-19 and thus driving these severe illnesses.

2) Further, pooling all coronaviruses is of questionable utility. Of even more questionable utility is the idea that these might represent a proxy for SARS-CoV-2 biology—the sentences that suggest this should be softened and revised thoroughly. It might be of some interest to break out the data from NL63 specifically since it uses the same receptor as SARS, but even then it is still biologically distinct. Even if the initial analysis and statistics is pooled, breaking them out in the supplemental data would be helpful

Response: Given the limited number of samples with CoV infections, pooling was the only option to run this analysis in a statistically powered fashion. We only have one sample with robust NL63 infection, precluding the proposed analysis. We acknowledge this pooling comes with limitations and as suggested we have softened conclusion statements related to the interpretation of our analyses, which are already significantly softened due to the revised negative result. Nonetheless, as detailed above we still believe this analysis is important to include in the manuscript.

3) Related to this, the abstract states that coronaviruses stimulate “a more pronounced cytotoxic immune response relative to other respiratory viruses”—this is not true, at best it is only relative to HRV. The language is similarly sloppy elsewhere and suggests a much larger conclusion than can be drawn (see also line 409).

Response: As described above in the revised analysis we no longer observe this enhanced CoV cytotoxic response, therefore this text has been removed.

4) The gene set enrichment analysis that underlies some of the “cytotoxic signature” presented in 6f and 6g do not look particularly convincing and no statistical significance is provided, between the shared and CoV specific signatures.

Response: All of these original results were statistically significant although p values were not reported in the figure, though they could be found in Supplemental Table 7 of the original manuscript. However, this is a moot point now as the revised analysis was not significant, and as such we no longer report his result in the paper and have accordingly removed the table in the revised version of the manuscript.

5) The introduction contains a sensational and overly earnest justification for the study. The statement “children are also thought to be highly susceptible to infection” is not cited and likely for good reason, as it is not supported by any reputable research. This is followed by the statement “Moreover, recent data suggest that 38% of COVID-19 cases occurring in children are of moderate severity and 5.8% are severe or critical”—this cites a single, poorly written and designed study which itself acknowledges that the 5.8% number comes from suspected cases rather than confirmed cases “There were more severe and critical cases in the suspected than confirmed category in this study. However, it remains to be determined if these severe and critical cases in the suspected group were caused by 2019-nCoV or other pathogens (eg, RSV). It may become clearer because the epidemic is quickly unfolding.” Further, their reported adult severity rate is significantly higher than rates reported in better designed and more reputable studies suggesting that this group’s selection criteria did not allow for a reasonable estimate of the infection fatality rate. There is no need for cherry-picking poorly designed studies that are completely out of line with the bulk of reported data to try to “juice” the significance of the current study.

Response: We appreciate the reviewer’s rigor and critical assessment of the cited study. However, the only “sensational” or unreferenced aspect of our introduction was that “children are highly susceptible to infection.” We have revised that phrase to accurately state, “children are also susceptible to infection.” Regarding the severity rate in children, we reported the only data we were aware of at the time, published in a highly respected journal, *Pediatrics*. However, we agree regarding the unconfirmed nature of cases, so we have added the caveat to their reported numbers of “...although not all cases were confirmed in this data set.” Secondly, we are now aware of childhood prevalence (2%) and severity

rate (5.7-20%) data from the CDC which we now report. We believe the revised paragraph is now balanced and not sensational.

6) This sentence seems tautological to me: line 102 “as many as 70% of genes expressed by a tissue or organ are under genetic control.”

Response: We agree with the reviewer. The sentence now reads “Indeed, expression quantitative trait loci (eQTL) analyses carried out in many tissues have suggested that the expression of as many as 70% of genes are influenced by genetic regulatory variants.”

Reviewer #2

This is a peer-review of paper by Sajuthi et al, which examined gene expression patterns of nasal epithelial cells obtained from brushes in over 500 children. They found that gene expression of ACE2, the putative receptor for SARS-CoV2, was significantly related to interferon response network and genes involved in the cytotoxic immune response; whereas TMPRSS2 was related to T2 mucus secretory network.

Major Comments:

1. One major limitation is that the investigators used nasal epithelia of children (who generally do not become sick with COVID19) rather than the more relevant adult tissues. Further, it would have been desirable to examine gene expression patterns in bronchial epithelia rather than nasal epithelia as the more severe COVID-19 cases involve lungs.

Response: While we acknowledge the perspective of the reviewer we would like to clarify a few points. First coronavirus infections including SARS-CoV-2 begin, and in many cases remain in the upper (nasal) airway. In fact, this is the most common body site to test for SARS-CoV-2 infection. Moreover, severity and other characteristics of upper airway infections may determine whether lower airway infection occurs. In this sense we believe study of the nasal airway is highly relevant to COVID-19 illnesses. Secondly, our study is primarily concerned with factors related to SARS-CoV-2 infection, at this point there is little indication that children are less susceptible to infection, but only that the likelihood of severe illness is significantly less. Thus, we believe that understanding factors related to childhood infections, which can then be spread to adults, who are more susceptible to severe COVID-19 illnesses, is very worthwhile. Additionally, across this epidemic a substantial number of significant COVID-19 illnesses will occur in children and reports are emerging even suggesting unique severe responses to COVID-19 illnesses in children (e.g. Kawasaki syndrome), therefore study of children is definitely warranted. Beyond this, we believe that many of the strong genetic and biologic factors affecting expression of these genes in children will behave similarly in adults and in bronchial epithelial tissue² (see study citing similarity in nasal and bronchial expression profiles).

2. There are no specific experiments using SARS-CoV2 and as such the data derived from other respiratory viruses are questioned for their relevance to COVID-19. Viruses behave differently and elicit different immune responses.

Response: Most experiments in the paper revolve around understanding genetic and biologic/inflammatory factors controlling population variation in the expression of the SARS-CoV-2 infection related genes. The premise and importance of looking at CoV infections relative to other virus species is explained in Reviewer #1, Response #1.

3. There are no protein data to corroborate gene expression information. Protein data would have strengthened the observation.

Response: We agree that protein data would indeed strengthen the gene expression data we have presented. As with point 1, this is a limitation we are forced to accept given the nature of the study being carried out retroactively on data from an existing cohort, nevertheless we still believe our study and results are important and impactful.

Detail Comments for each section are denoted below:

Introduction

Page 6. Line 103-105. This seems that is not necessarily connected to SARS coronaviruses and host genetics. How is HRV similar to SARS-CoV-2? Are there any specific similarities? Both viruses have S proteins? or part of the genome is comparable?

Response: The variants in genes *CDHR3* and *ORMDL3* are not necessarily connected to the infectivity SARS coronaviruses. Rather, these are concrete examples from the literature of genetic variants influencing the severity of specific viral respiratory illnesses^{3,4}, providing a basis for considering genetic variants as risk factors for SARS-CoV-2 infection and COVID19 outcome.

Page 6. Line 105-107. The existence of genetic polymorphisms is only relevant in presence of phenotype variability. Given the phenotype variability of COVID-19 in children and the presence of genetic variations in the population (even within in cohorts of similar individuals), then it is possible that functional variants could explain differences in severity. I suggest reviewing this justification.

Response: See above response. We agree with the reviewer's point regarding the relevance of polymorphisms depending on their association with phenotype variability. Our paper performs a full genome sequencing based cis-eQTL analysis (in the airway) of genes highly involved in SARS-CoV-2 infections. The eQTL variants identified are prime candidates for influencing phenotypes of SARS-CoV-2 infection and COVID-19 illness severity, and provide focused targets for future studies evaluating genetic influence on variability in COVID-19 severity.

Page 6. Line 114-119. I suggest removing this.

Response: We respectfully disagree, see justification above. Moreover, this section of the introduction highlights that we can use the data from our system to provide some insight into the transcriptional immune response in the airway to human coronaviruses infections. This baseline will be valuable as a reference for the unique and unusual immune reaction observed in response to SARS-CoV-2.

Methods

General remarks: Overall methods are well described; however, the organization could be improved. A concise version of the methods in the main manuscript would be beneficial to the reader. Specific details can be included in the supplementary files.

The authors should correctly refer to the abbreviations, additionally, sample sizes use for each analysis should be clear from the beginning.

Response: We thank you for these suggestions. Abbreviations and descriptions have been corrected throughout. Sample sizes have been clarified throughout the methods. We will follow the editor's direction with regard to the level of detail to be included in the methods in the body of the text vs. in the supplement.

Page 25. Line 551 - 552. Please provide here the number of individuals used from the GALA cohort (line 551 page 25) and the number of subjects used use for ALI.

Response: The sample sizes for the network (n=695), eQTL (n=681), and ALI (n=5) analyses have been included, as requested.

Page 26. Line 574. Which measures of spirometry were obtained used for this study? Were they used in any way?

Response: No measures of spirometry were used in this particular study. The text now reads "Additional physical measurements were obtained, and subjects provided a blood sample for DNA extraction".

Page 26. Line 575. Provide abbreviation for Whole Genome Sequencing when it is first introduced.

Response: Abbreviations and descriptions have been corrected throughout.

Page 27. Line 583. Describe the meaning of the abbreviations (i.e.: ALI and WGS) when they are first introduced.

Response: Abbreviations and descriptions have been corrected throughout.

Page 27. Line 591. How many ALI samples were BULK RNAseq? All GALA cohort or 5 subjects? Please clarify and provide n from the beginning.

Response: Bulk RNA sequencing was carried out on nasal brushings from 695 GALA II participants, as well as the ALI samples from 5 subjects. The text has been edited to include these sample sizes in this section.

Page 28. Line 607 - 608. I suggest including a small summary of the genotype by sequence approach used.

Response: We now include the following summary:

"Briefly, the sequences were mapped using BWA-MEM to the hs38DH 1000 Genomes build 38 human genome reference. Variants were identified and called using the GotCloud pipeline. These variants were filtered for quality, and genotypes with at least 10X coverage were phased with Eagle 2.4."

Page 28. Line 612 - 613. ALI description protocol and culture should be provided earlier in the methods when the concept is first introduced.

Response: The ALI protocol has been moved earlier, subsequent to the description of the GALA II cohort from which the ALI samples were derived.

Page 28. Line 616 - 617. Was there a criterion used to select these 5 subjects? If so, please mention it.

Response: No selection criteria was used to obtain these subjects, they were selected randomly from the pool of deidentified sample labels. The manuscript text has been updated to reflect this, by including the phrase “from five randomly selected GALA II subjects”.

Page 31. Line 575 - 576. How did the authors determine which genes ‘underlie’ T2 and interferon inflammation? Were only these genes used for the hierarchically clustered of the subjects? Explain.

Response: The T2 and interferon inflammation networks were determined as such using both Gene Ontology enrichment terms obtained for these networks, cell type enrichments, as well as confirming that they included genes known to be involved in this type of inflammation. For example the top enrichment for the T2 network (saddlebrown) was “IL-13 human airway epithelial cells.” Other top terms included “neurotrophin binding” which in the nasal epithelia has been associated with Type 2 inflammation⁵, as well as “extracellular matrix organization” which is consistent with the abundant tissue remodeling observed in type 2 high individuals⁶. Moreover, the member genes of the saddlebrown module included a range of top reported markers for type 2 inflammation including CLCA1, CCL26, and POSTN⁷.

The enriched Terms enriched for the Interferon (tan) network include an array of Interferon related ontologies, such as “Type I interferon signaling” and “Cellular response to type I interferon”, characteristic of an interferon response to viral infection⁸. The member genes of the tan network include genes known to be upregulated in virally infected individuals including IFNG, CXCL11, and IFIT3⁹⁻¹¹. Therefore, we feel extremely confident in our designations. For reference select enrichment terms and network hub genes are shown in Figures 2a (T2: saddle brown) and 4c (Interferon: tan) and complete enrichment data is now shown in Supplementary Table 2. Also please see our related response to comment #3 of Reviewer #3.

Secondly, the reviewer is correct that only the network genes were used to hierarchically cluster the subjects for assignment into T2 High vs Low (saddle brown) or Interferon high vs low (tan). The text has been updated to reflect this more specifically.

Page 32. Line 700 - 703. What was the p value threshold for the eQTL analysis? Considering the sample size and MAF, I wonder if you should assess the rare variant with a more stringent p value? I don't think this will change the bottom line of the manuscript. Also, did the authors set a LD window and MAF for the stepwise regression analysis? Please include this information.

Response: We generally followed the GTEx consortium pipeline for cis-eQTL analysis including significance threshold selection, selection of other parameters, and in the general methods used. This info is listed below and is now in the manuscript methods. Regarding rare variant analysis, we already analyze variants with a frequency as low as 1%, testing of variants with frequencies less than 1% without

a replication cohort would be of limited value due to uncertainty in the finding, therefore we feel a 1% cutoff for testing is appropriate.

“To perform cis-eQTL analysis, we utilized a modified version of FastQTL that was provided by the GTEx project, with a MAF cutoff of 1% and a cis window of 1Mb. The results from FastQTL’s permutation pass were used to generate a genome-wide empirical p-value threshold for each gene, according to the GTEx method. In short, the empirical p-values were adjusted for multiple testing, and the p-value closest to the FDR threshold of 0.05 was used to compute a nominal threshold for each gene based on the modeled beta distribution from FastQTL. These gene-level significance threshold ranged from 1e-6 to 1e-2.”

Page 33. Line 721. Describe abbreviation of CoV

Response: Abbreviations and descriptions have been corrected throughout.

Page 33. Line 741. “The function arrayWeights function” needs to be corrected.

Response: The text now reads “the arrayWeights function”.

Page 34. Line 757 - 760. “For cell type enrichments within WGCNA networks reported in Figure 2a, we tested for overrepresentation of network genes within gene marker sets (FDR < 0.05) for each of 35 epithelial and immune cell types inferred using scRNA-seq of human lung tissue.”

This sentence is too complicated and hard to follow, please re-write. In addition, the authors should provide a brief description of the scRNA-seq of human lung tissue used to infer the epithelial and immune cell types.

Response: The section has been rewritten, identifying the source of the gene markers and clarifying the sentence structure. The revised text is included below:

“To determine if WGCNA network genes were enriched for markers of particular cell types we performed an enrichment analysis leveraging a published scRNA-seq data set which defined marker gene lists for 35 epithelial and immune cell types within lung tissue¹². Specifically, the WGCNA network genes (Figure 2a) were tested for overrepresentation within each cell type marker gene list, using an FDR cutoff of 0.05.”

Page 34 - 35. Line 765 - 766. Are the authors referring to testing differential expression between immune cells and the rest of the cell types? How many cell types (other than immune) were in the data?

Response: The differential expression was carried out between each immune cell type vs the remaining 10 immune cell types to create reference pre-ranked gene sets for GSEA. No additional cell types were included in creating those reference gene sets. The text has been edited to read “limma was then used to perform differential expression analysis between each immune cell type and the remaining 10” to clarify this approach.

Page 37. Line 815 - 816. Is the title “Analysis of bulk RNA-seq data from IL-13 and HRV infected ALI nasal airway epithelial cultures”, refers to “IL13-stimulated and HRV infected”?

Response: That is correct. The title has been edited as suggested.

Page 38. Line 857. CODE AVAILABILITY information is missing, please provide repository use to deposit the codes.

Response: The scripts used to perform the analyses described in the paper have been deposited to the GitHub repository: https://github.com/seiboldlab/ACE2_TMPRSS2_Airway

Results

General remarks: The authors aimed to fully described the results of this study. However, in multiple instances the sentences were too long and wordy, particularly at the beginning of this section. A study demographic table should be provided.

Response: As requested a demographic table on the GALA II study has been provided as Supplementary Table 8.

Page 9. Line 173 - 174. Are these the top expression programs represented by the networks?

Response: They are among the enriched terms and representative of the top enrichments. We have revised enrichments reported in the figures and text to increase clarity and solidify our points, justified the terms selected in the Reviewer #3 Response #3, and also in response to your comment above, and reported the full list of significant enrichments in Supplementary Table 2.

Page 9. Line 176 - 185. Run-on sentences. I suggest providing one clear example instead of these three.

Response: We feel that all three examples discussed here are relevant in their associations with TMPRSS2 expression, and text provides helpful context for the data included in Figure 2. However, we have revised the sentence structure make this clearer.

Page 10. Line 204-210. Although figure 3 looks convincing; It is a strong statement to say that this experiment can assess causality between T2 inflammation and TMPRSS2/ACE2 expression.

Response: In the referenced experiment we directly test, *in vitro*, if IL-13 alters TMPRSS2/ACE2 expression, which it does, moreover this finding provides a clear explanation for the *in vivo* results. We stand by our statement.

Page 12. Line 247. "Metagenomically" is not the correct way to express genome-wide meta-analysis. Please correct this.

Response: The word "metagenomically" in the text refers to the method of analyzing both the host and respiratory virus genomes together from a single RNA-seq sample.

Page 12. Line 260 - 261. Sure, but only HRV-A viral infections, the effect of other viruses remains to be tested. Please clarify that the results are specific to the virus been used in the experiment.

Response: The network data suggests this, and we now show ACE2 is upregulated by CoV infections and other respiratory virus infections in our *in vivo* infection data (New Figure 6f), in the later section. But we now temper the statement at this place in the manuscript.

Page 13. Line 269 -270. What p threshold was used to identify associated variants?

Response: We performed cis-eQTL analysis according to the GTEx_v7 pipeline as summarized in the methods section. The p-value threshold to determine significant eQTL variants for ACE2 and TMPRSS2 is $1e-5$ and $3e-5$ respectively. For each gene, the nominal p-value threshold is a function of the number of tested variants within a 1Mb cis-window.

Page 13 - 14. Line 283 - 298. This looks more like interpretation rather than results. I suggest removing these lines from the results.

Also, different AF in different population don't suggest that if TMPRSS2 levels influence susceptibility to SARS-CoV-2, then genetics may play a significant role in infection risk. Different population may have different AF of the variants simply because of genetic drift or population structure. Without more supporting information on other cohorts this is not a correct statement.

Instead, the results suggest that even within a cohort similar ancestry, expression vary due to eQTLs (at least partially), hence genetic determinants of TMPRSS2 expression may play a significant role in infection risk.

Response: We believe it is standard practice in genetics manuscripts to describe the allele frequencies of discovered functional variants in world populations. Moreover, we believe the readers will be very interested in this data, therefore we describe this data in the results and display in the figure. Regarding our interpretation, since on average eQTL effects are very consistent across population¹³ and we have shown that: (1) significant eQTL variants exist for *TMPRSS2* in the airway, and (2) that the frequency of these eQTL variants vary across world populations, it follows as we state in the manuscript that, "if *TMPRSS2* levels influence susceptibility to SARS-CoV-2, then genetics may play a significant role in infection risk and that this risk will vary significantly across world populations." This is a speculation, but the logic behind the speculation is accurate. As a speculation, we have moved this sentence to the discussion section.

Page 15. Line 314 - 316. "While the ACE2 eQTL variant, rs181603331, was associated with a notable decrease in ACE2 levels, it only accounted for 1.2% of the variance, reflecting the low frequency of this variant in our population." How do the authors reach this conclusion without calculating the additive variant component?

It is more likely that the effect of the single SNP is inflated due to its low MAF. Unlike common variants, rare ones have a larger effect size on a trait. The heritability of the expression is not presented here; therefore, you don't know how much of the expression is due to overall genetics, or whether if 1.2% for a single SNP can be considered a small proportion of what can be capture from the heritability. Please review your interpretation and move it to the discussion.

Response: The point of the multi-variable analysis for ACE2 expression in section 5 of the paper is both to determine the effect size of the tested variables (eQTL variants and inflammatory endotype), while accounting for all variables, and also to determine the percent of overall variation in expression accounted for by each variable. In multi-variable regression modeling the percent variation accounted for by each variable is generated by the partial r^2 statistic, as we report in paper Table 1. As you can see from the beta coefficient for rs181603331 the effect size is rather large; it is over 50% of the effect size for T2 inflammation, which has a strong effect on ACE2 levels. However, the variation accounted for by this variant (partial $r^2=1.2\%$) is only 5% of the variation accounted for by T2 inflammation (partial

$r^2=24\%$). This phenomenon is certainly related to the fact that only 27 of 681 subjects harbor the rs18160331 minor allele, limiting its explanatory power for population-wide variance in expression. Lastly, regarding an inflated effect size for the rs18160331, while we agree that estimating effects with small numbers of samples is subject to unstable estimates, we don't believe this is the case here. We have a reasonable number of subjects with a minor genotype for this SNP to estimate effects, and the FDR for this eQTL is $6.3e-18$. The cis-heritability estimate for ACE2 expression level is 3% (computed using GCTA on 1MB window from the transcription start site of ACE2).

Page 18. Line 386-387. I really like the comparison between CoV and HRV. However, the statement at the end "These results suggest that while CoV infections are highly similar to HRV infections, they likely elicit an enhanced cytotoxic immune response" seems a bit strong, I suggest been cautious when referring to the similarities between type of viruses, particularly in the result section.

Response: The reviewer makes a good point about the limitations of our findings and in fact the revised analysis has proved our comment to be an overinterpretation. The new analysis comparing CoV to multiple other viruses no longer produces this effect, and we have been careful to not overinterpret the revised analysis as well.

Discussion

General remarks: Limitations of this study are not fully stated in the discussion. HRV infection experiments results and its implications with COVID-19 infection could be expanded.

Response: We have expanded limitations and discussion of virus comparison analysis as suggested.

Page 19, Line 396 - 405. I suggest reviewing and summarizing this sentence.

Response: We have modified.

Page 19. Line 414 - 426. The main focus of this study are children, however the scRNAseq was obtained from a middle-aged adult. It is quite possible that expression changes with age. How do you think this difference could affect your conclusions.? You should mention this limitation. In addition, authors should acknowledge that scRNAseq from one subject may not be representative of the population.

Response: We simply use this analysis to establish cell types expressing these genes in the nasal epithelium. No other results in the paper are dependent on these findings. Our results are in line with other more extensive scRNA-seq papers analyzing these genes. We now acknowledge limitations in the paper as suggested.

Page 20. Line 442 - 449. I suggest reviewing and summarizing this sentence. What is the meaning of "complicating expectations" in this context?

Response: We hypothesize that expression levels of TMPRSS2 and ACE2 may influence risk for a poor COVID-19 outcome. T2 inflammation leads to upregulation of TMPRSS2, and downregulation of ACE2. These opposite trends of gene expression make it difficult to conclude the impact of T2 inflammation on COVID-19 outcome directly. We have revised these lines to make clearer.

Page 21. Line 449 - 456. Are there other signs of T2 inflammation in severe or fatal COVID-19 cases that could further support your hypothesis?

Response: As referenced in the introduction, it was reported that in a cross-sectional study of fatal COVID-19 patients that they exhibited very low blood eosinophils, possibly suggesting T2-low subjects may have increased susceptibility to poor COVID-19 outcomes. However, this association could also be a result of having the illness. We are not aware at this time of other studies investigating associations between T2 inflammation and severity of COVID-19 illness.

Page 23. Line 491 - 489. Why did the authors stated that variants may have strong effects sizes in a larger population? This is not likely. Overall, as a study cohort gets larger, effect sizes will become smaller because of a wider range of phenotypes. Please review your statement. eQTLs are very common across the genome, however only few eQTLs were identified on the GALA cohort, which suggest genetics have a small influence on the natural variation of expression. However, it is also possible that the eQTL analysis was underpower, therefore missing relevant eQTLs. This limitation needs to be acknowledged by the authors.

Response: It was not our intent to state that the variants may have a stronger effect in a larger population. Rather, our intended meaning was in populations where the eQTL variants are more common, the variant could account for more of the population variance in expression of these genes. Secondly, if expression levels of these genes influence infection rate and severity, then it follows that these variants would be associated with infection rate and COVID-19 severity. We have revised to make this point clearer.

“More frequent variants” are called common variants. Please change.

Response: We have changed.

Reviewer #3

The manuscript describes expression of ACE2/TMPRSS2, two SARS-CoV-2 related genes, in human nasal epithelium and predicts potential mechanisms regulating their expression. The authors mine next-generation sequencing data (bulk and single cell RNA-seq, whole genome sequencing, cis-eQTL analysis, metagenomic analysis), to demonstrate the cellular expression pattern of ACE2/TMPRSS2 in nasal epithelial cells and immune sub-types. They identified a number of factors associated with ACE2 or TMPRSS2 expression in nasal epithelium, including type 2 inflammation, “interferon” response, eQTL variants, and respiratory virus infection. Using multivariate regression analysis, the authors estimate the relative importance of these factors to ACE2/TMPRSS2 expression. While of interest, relationships among QTL, ACE2/TMPR and patient endotypes are not identified in the present work. Novelty of the present work may be impacted by several recent publications, see (e.g., Sungnak et al., Nat Med 2020, Ziegler et al., Cell 2020, Jackson et al., JACI 2020, etc.) who identified a number of the findings presented in this paper, including the associations of type 2 inflammation, type 1 interferon, and asthma with ACE2/TMPRSS2 expression in nasal epithelium.

Response: Thank you for your thorough review of our paper. We disagree that “relationships among QTL, ACE2/TMPRSS2 and patient endotypes are not identified in the present work,” in fact that is exactly what we have done: examining co-expression networks representing endotypes (T2-high and Interferon-high), determining their relationship with ACE2 and TMPRSS2 expression. Moreover, in section 5 of the paper we examine all relevant factors: QTLs, endotypes, asthma status, and demographic variables in determination of population variation of ACE2 and TMPRSS2 levels.

Regarding novelty, the Sungnak and Ziegler papers (Dr. Seibold is a co-author on both as a member of the HCA Lung Biological Network) are highly focused on defining cell types that express ACE2 and TMPRSS2, which is a very minor component of our manuscript. The Jackson letter simply reports a relationship between allergic traits in asthmatics and ACE2. The biggest point of overlap is with the Ziegler paper that reports ACE2 as being induced by interferon signaling. However, as noted above, the core of our paper is fundamentally different in examining in vivo nasal epithelial expression data and whole genome sequencing data in a large sample (n>600) of children with network, eQTL, and multivariable modeling allowing us to:

1. Define all genetic variants influencing expression of ACE2 and TMPRSS2 in the nasal airway
2. Define the most important biological networks that control ACE2 and TMPRSS2 expression in the airway
3. Find that the biological networks that control ACE2 and TMPRSS2 expression and define inflammatory endotypic groups
4. Determine which of these factors are most important in determining population airway expression levels of the ACE2 and TMPRSS2 genes.

Lastly, we note our paper was submitted near simultaneously to the referenced papers, however the review process has been longer for our paper, so we don't believe that should be held against our submission.

Specific Comments:

1. 10X 3' v3 scRNA-seq of nasal brushing from a single asthmatic subject is provided. Single cell expression of ACE2/TMPRSS2 in nasal cells from healthy donors and those with allergic inflammation were reported in several studies (e.g., Sungnak et al., Nat Med 2020; Ziegler et al., Cell 2020). These studies also showed cellular expression patterns of ACE2 in scRNA-seq data from in vitro nasal epithelial cell cultures. **Insights added by the scRNA-seq from a single asthmatic subject in the present work are unclear.**

Response: We are familiar with these papers and in fact Dr. Seibold is a co-author on these publications as a Human Cell Atlas contributor. The single cell data presented in our paper is a minor, ancillary portion of the paper, but serves to orient the reader as to the cell type expression of these genes in the nasal epithelium of an asthmatic, forming some level of expectation for the bulk RNA-seq expression of these genes, which is a central part of the paper. Moreover, neither of the referenced papers contain data from asthmatics which are known to exhibit a highly remodeled airway epithelium, so our data is novel in that respect. Finally, that our results largely mirrored the HCA papers is valuable given the reproducibility crisis in research and the importance of having confidence in findings related to understanding a worldwide pandemic. Therefore, we believe this section is still valuable to include in the paper.

2. The authors used scRNA-seq data to identify cell types. **The authors should specify the markers they used to define cells. What defines and distinguishes different secretory cells?** A two-stage analytic approach was used to identify cell types. In the first stage, cells passed a QC filtering and clustering analysis identified epithelial or immune lineages of the cells. A second cell filtering was used to remove about 3,000 (~ 30%) epithelial cells. **The reason for this second filtering was not well explained.** It is not clear whether the expression of ACE2 or TMPRSS2 in those “removed” epithelial cells affected the overall patterns?

Response: We apologize for the oversight of not including the markers for the scRNA-seq data cell clusters (types), this information is now provided in Supplementary Table 9 (in vivo) and 10 (in vitro). As others have reported, agnostic clustering airway epithelial cells identifies a spectrum of secretory cell types, including differentiation intermediates and functional subtypes, all of which are poorly understood. We label these cell types as best we can and we now provide pairwise DEGs for these secretory clusters (Supplementary Table 9 and 10) so readers can make their own determination of the function/nature of each cluster.

Regarding the filtering strategy: this was explained in methods section, “Analysis of scRNA-seq data from the nasal epithelial brushing,” however, we add more detail here. Since we have found in this dataset and in other cellularly heterogeneous tissues that QC metrics and gene counts can vary greatly between different cell types, we usually apply liberal QC cutoffs for our tissue scRNA-seq datasets, cluster, determine general cell populations and then apply more conservative cutoffs that are appropriate for disparate groups of cells. In this case as in other datasets we find the number of expressed genes for many immune cell types is much less than epithelial cells. Therefore, we applied a different gene count cutoff for epithelial cells. Examining the gene count distribution for epithelial cells notice it has a bimodal appearance, breaking at ~1,000 genes. Moreover, if you look at the gene counts for the clusters identified by the first stage below, you will find that 2 epithelial clusters contain the vast majority of cells with less than 1,000 genes. The marker genes for one of these clusters is mostly mitochondrial genes and for the other cluster is mostly ribosomal genes, signaling that these are low quality cells. Therefore, we felt justified in using this gene number cut. Neither of these clusters were remarkable with respect to ACE2 or TMPRSS2 expression, as shown below, and therefore this cut has not affected our results or interpretation.

3. The authors used gene co-expression network and hierarchical clustering of bulk RNA-seq from nasal brushings from 695 healthy and asthma donors (children of Latino descent in Puerto Rico, 331 control and 364 donors with asthma), to identify an in vivo relationships between airway type 2 inflammation and TMPRSS2/ACE2: samples with high T2 inflammation showed high TMPRSS2 but low ACE2 expression. TMPRSS2 was highly correlated with a gene set enriched for goblet cell markers and T2 cytokines IL13; ACE2 expression was negatively correlated with this gene set. Using in vitro stimulation of paired air-liquid interface (ALI) airway cultures, the authors showed that IL13 increased TMPRSS2 and decreased ACE2. Using scRNA-seq data from tracheal epithelial cultures that were chronically treated (10 days) with IL-13 or control media, the authors showed cell specific patterns of TMPRSS2 and ACE2 regulation by IL-13. The pathway and cell type enrichment analysis of the identified gene sets is important in this analysis. **For each gene set, only one or two selected pathways or cell types were listed. Were the listed pathways/cell types the most enriched ones to best characterize each gene set?**

Response: We appreciate the reviewers concern here whether the enrichments presented in the paper best characterize the networks. We also ask the reviewer to recognize that in this analysis we are testing hundreds of thousands of terms from multiple gene ontology databases. As a result, many of the terms between databases are redundant or describe in different ways the same processes. For example, the number 1 and number 3 enrichments (by adjusted p value) for the pink network genes are “Interleukin-13 (IL-13) human airway epithelial cells (AECs)” and “interleukin-13 human esophageal epithelial cells”, respectively, both of which heavily overlap in the genes these terms contain. But the essence of both terms is they represent IL-13 response genes, so in the main text of the paper there would be no need to report both, but rather to report a term like the 4th most significant one, “Mucin type O-glycan biosynthesis” that transmits to the reader a biological process, mucus generation, that is encompassed

by the network. Additionally, many enriched terms are so generic in term name that they add little understanding to a set of genes, for example terms like “immune system.” Therefore, in the paper figure we tried to select terms for display that were among the top ranked, highly statistically significant, non-redundant, and biologically informative. In fact, with the reviewers comment here we have examined this further and revised displayed terms to even better meet these goals. We list the revised terms in the figures and here, which have not changed any interpretations of the networks or conclusions in the paper. We also include the complete list of significant enrichment terms in Supplementary Table 2. From these complete lists you will also see our selection of terms for the figure and description of function in the results text is accurate. Also see our response above to Reviewer 2 that further justifies our description of the networks.

Network	Adjusted P Value Rank	GO Term	Adjusted.Pval
Black	1	Asparagine N-linked glycosylation Homo sapiens R-HSA-446203	1.93937E-14
Black	2	Transport to the Golgi and subsequent modification Homo sapiens R-HSA-948021	6.77341E-10
Black	3	Interleukin-13 (IL-13) human airway epithelial cells (AECs) GDS4981 ligand:72	3.67049E-09
Black	12	COPI-mediated anterograde transport Homo sapiens R-HSA-6807878	9.40906E-06
Pink	1	Interleukin-13 (IL-13) human airway epithelial cells (AECs) GDS4981 ligand:72	3.03887E-47
Pink	4	Mucin type O-glycan biosynthesis	8.89505E-05
Pink	5	O-linked glycosylation of mucins Homo sapiens R-HSA-913709	0.000326811
Saddlebrown	1	Interleukin-13 (IL-13) human airway epithelial cells (AECs) GDS4981 ligand:72	8.52057E-29
Saddlebrown	2	interleukin-4 human keratinocyte GDS4601 ligand:178	0.017923983
Purple	1	Immunoregulatory interactions between a Lymphoid and a non-Lymphoid cell Homo sapiens R-HSA-198933	8.97654E-22
Purple	9	Adaptive Immune System Homo sapiens R-HSA-1280218	8.79026E-16
Purple	28	Natural killer cell mediated cytotoxicity	8.59474E-11
Tan	1	interferon-I +/- human hepatocyte GDS4390 ligand:179	7.9276E-149
Tan	8	Interferon Signaling Homo sapiens R-HSA-913531	1.56379E-44
Tan	12	type I interferon signaling pathway (GO:0060337)	1.39033E-34

4. The gene co-expression network identified ACE2 as highly correlated with that of a gene set enriched for interferon and viral responses. Hierarchical clustering analysis using this “interferon” gene set partitioned samples into “interferon-high” and “interferon-low” groups. ACE2 expression was increased in the “interferon-high” group vs. “interferon-low”. The “interferon-high” and “interferon-low” groups were defined on the expression of a de novo gene set which was predicted to be associated with interferon responses based on pathway enrichment analysis. **The true interferon status of the subjects in these two groups merits an independent validation.** A recent study demonstrated decreased nasal epithelial expression of ACE2 in children with asthma (Jackson et al., JACI 2020). The present analysis used data from both healthy and asthmatic subjects, thus, **it is not clear what the distribution of asthmatic subjects was in the “interferon-high” and “interferon-low” groups, and the impact of asthma status on the findings was not entirely clear.**

Response: To validate that the interferon assignment is correct we show mean expression of genes from the gene ontology term “interferon signaling”, in subjects assigned to the interferon-high vs. interferon-

low groups. As you can see from the figure below, the interferon status assignment is excellent. Also see our response to reviewer 2 above.

Regarding the effect of asthma status reported by Jackson et al., we strongly believe this effect is due to the high prevalence of T2 inflammation that will be present in any asthma cohort. In fact, both our work and theirs clearly shows this is the case. Reinforcing this we show that when T2 status is accounted for in the multivariable model, the ACE2 association with asthma is no longer seen. To demonstrate this further, we show ACE2 expression by both asthma and T2 status.

The interferon effect on ACE2 levels is also independent of asthma status as is seen in the figure below. Moreover, the frequency of interferon high subjects is not statistically different between asthmatics and healthy subjects.

5. Using scRNA-seq data from in vitro epithelial cultures, the authors concluded that expression patterns of ACE2/TMPRSS2 were similar to those from “in vivo” nasal scRNA-seq data. However, in vivo scRNA-seq was represented by nasal brushing from a single 56-year-old asthmatic donor and the in vitro scRNA-seq used tracheal epithelial cells from a single de-identified lung donor (the lung condition and age information were unknown). **Two different methods (from pre-processing to clustering to differential expression) were used to analyze these two scRNA-seq data. Thus, there are concerns regarding the cell type assignments in these distinct tissues.**

Response: Limited conclusions were drawn from the in vivo scRNA-seq data, namely that both ACE2 and TMPRSS2 were expressed across epithelial cell types, with higher expression of TMPRSS2. These results are in line with the other single cell papers discussed here.

The in vitro scRNA-seq was used to support the bulk RNA-seq network response results and give single cell explanatory power to the in vitro bulk RNA-seq stimulation results.

The only conclusion drawn in the paper between in vitro and in vivo scRNA-seq data is that most all epithelial cell types express both ACE2 and TMPRSS2 and that TMPRSS2 is expressed at a higher level. The minimal, different processing of the in vivo and in vitro scRNA-seq data would not affect this very limited and minor paper conclusion, which is also supported by the much larger scRNA-seq data sets recently published and discussed here.

6. Using cis-eQTL analysis of bulk RNA-seq of nasal airway epithelial brushings and whole-genome sequencing of 681 subjects, the authors identified eQTL variants associated with ACE2 or TMPRSS2 expression. The frequencies of those eQTL variants among eight different populations were investigated using the public gnomAD genetic variation database. The authors statement that “if TMPRSS2 levels influence susceptibility to SARS-CoV-2, genetics may play a significant role in infection risk and that this risk will vary significantly across world populations” **is highly speculative. It is presently unclear whether ACE2/TMPRSS2 levels influence COVID infection or severity.** The present eQTL variants were identified using data from both asthma and healthy donors, thus, **it is unclear what the impact of asthma on expression is.** Associations were identified using data from a single population (children of Latino descent between the ages of 8 and 21 in Puerto Rico) and thus, **merits validation in a separate cohort.**

Response: To be clear we show a statistically significant TMPRSS2 eQTL (rs1475908, $p=3.8e-11$) in our population. Moreover, the eQTL analysis included asthma status as a covariate and running the analysis only on healthy control subjects only also find this SNP as a TMPRSS2 eQTL ($P=3.6 \times 10^{-3}$). We also note our analysis used whole genome sequence data and we performed an analysis to reduce down all associated SNPs to the likely causative SNP or at least the causative SNP block. Therefore, we don't believe it is “highly speculative” to assume the SNP will behave as an eQTL in other populations, where it does exist, rather we suggest it is more speculative to assume the SNP will not behave as an eQTL in another human population without evidence, please see reference ¹³. However, we now a comment to the discussion that the eQTLs will ultimately need to be validated in other populations. As the reviewer notes, our speculation extended to the possibility that these TMPRSS2 eQTLs could influence COVID infection and severity, which we clearly premised on the completely reasonable idea that TMPRSS2 expression levels could influence COVID infection and severity. However, since this is speculation we have moved our comment to the discussion section.

In summary we strongly stand by our reasonable speculation which brings to light the significance of our work. In fact, we suspect and hope that genetic researchers will leverage our eQTL data to test this hypothesis directly in COVID cohorts.

7. Using metagenomic analysis of the nasal epithelial brushings bulk RNA-seq data from healthy and asthmatic children, the authors identified subjects harboring reads for a respiratory virus. “To discriminate CoV-enhanced responses from those that are more general to respiratory viruses”, the authors compared the transcriptomic profiles of subjects likely infected with a coronavirus with the profiles of subjects likely infected with human rhinovirus species, and those not infected with a virus. The metagenomic analysis also identified a comparable number of subjects likely infected with influenza. **Inclusion the influenza group for comparison might improve identification of potential CoV enhanced or specific responses.**

The authors state that “even asymptomatic carriage of respiratory viruses exacts a fundamental change in airway epithelial expression and immune cell presence, including upregulation of ACE2 expression”. **Nevertheless, in the present multivariate regression, this factor is not included. What is the relative importance of asymptomatic carriage of virus on ACE2 compared to other factors?**

Response: Regarding inclusion of influenza infected samples for the comparator group, this has been done. See extensive discussion in response to Reviewer #1 Response #1.

Regarding inclusion of virus infection as a covariate in the ACE2 expression multivariate regression model: we show in the manuscript that virus infection stimulates a host interferon/anti-viral response network, which ACE2 is part of. So we believe the order of causation is virus infection triggers the interferon network, that results in an increase in ACE2 expression. This idea is buttressed by the recent work by Ziegler et al, 2020 suggesting ACE2 is an interferon stimulated gene. We also know that not all virus infected people exhibit an interferon response (although 57% do in our asymptomatic dataset) and that non-viral infected people can exhibit high interferon signaling for other reasons. Therefore, we include the directly causative, thus more explanatory variable in the model (interferon status).

This is reinforced by the plot of ACE2 levels in GALA stratifying subjects by a combination of interferon status and viral infection status.

Minor comments:

(1) In line 373 of page 17, citation is missing.

(2) There are missing symbols in Methods, such as “Trans-Omics” on page 27

Response: We appreciate the reviewer’s sharp eye and attention to detail. These issues have been addressed in the text.

1. Lambert, K.A. *et al.* The role of human rhinovirus (HRV) species on asthma exacerbation severity in children and adolescents. *J Asthma* **55**, 596-602 (2018).
2. Poole, A. *et al.* Dissecting childhood asthma with nasal transcriptomics distinguishes subphenotypes of disease. *J Allergy Clin Immunol* **133**, 670-678 e612 (2014).
3. Everman, J.L. *et al.* Functional genomics of CDHR3 confirms its role in HRV-C infection and childhood asthma exacerbations. *Journal of Allergy and Clinical Immunology* **144**, 962-971 (2019).
4. Caliskan, M. *et al.* Rhinovirus wheezing illness and genetic risk of childhood-onset asthma. *New England Journal of Medicine* **368**, 1398-1407 (2013).
5. Watanabe, T. *et al.* Brain-Derived Neurotrophic Factor Expression in Asthma. Association with Severity and Type 2 Inflammatory Processes. *Am J Respir Cell Mol Biol* **53**, 844-852 (2015).
6. Dunican, E.M. & Fahy, J.V. The Role of Type 2 Inflammation in the Pathogenesis of Asthma Exacerbations. *Ann Am Thorac Soc* **12 Suppl 2**, S144-149 (2015).
7. Woodruff, P.G. *et al.* T-helper Type 2-driven inflammation defines major subphenotypes of asthma. *American Journal of Respiratory and Critical Care Medicine* **180**, 388-395 (2009).
8. Mesev, E.V., LeDesma, R.A. & Ploss, A. Decoding type I and III interferon signalling during viral infection. *Nat Microbiol* **4**, 914-924 (2019).
9. Kang, S., Brown, H.M. & Hwang, S. Direct Antiviral Mechanisms of Interferon-Gamma. *Immune Netw* **18**, e33 (2018).
10. McErlean, P. *et al.* Asthmatics with exacerbation during acute respiratory illness exhibit unique transcriptional signatures within the nasal mucosa. *Genome Med* **6**, 1 (2014).
11. Wesolowska-Andersen, A. *et al.* Dual RNA-seq reveals viral infections in asthmatic children without respiratory illness which are associated with changes in the airway transcriptome. *Genome Biology* **18**, 1:17 (2017).
12. Travaglini, K.J. *et al.* A molecular cell atlas of the human lung from single cell RNA sequencing. *bioRxiv* (2020).
13. Wen, X., Luca, F. & Pique-Regi, R. Cross-population joint analysis of eQTLs: fine mapping and functional annotation. *PLoS Genet* **11**, e1005176 (2015).

REVIEWERS' COMMENTS:

Reviewer #1 (Remarks to the Author):

I appreciate the authors' thorough response to the review and think that the revision to include comparisons to a broader "other respiratory virus" group significantly enhances the analysis. The fact that it is "negative data" in no way detracts from its value to the field--I think this will be a very useful resource.

The other edits address my primary concerns.

Reviewer #2 (Remarks to the Author):

The revised manuscript is improved, though not sufficiently. The key points regarding have not been fully addressed:

1. transcriptional comparison of "all coronaviruses" vs. rhinoviruses was largely dismissed. Viruses are unique and elicit specific responses. Pooling of all coronarviruses really misses the mark.
2. there are no protein data in the original manuscript and none in the revision. This reduces the impact of the paper.

Reviewer #3 (Remarks to the Author):

The authors have carefully considered the comments from the reviewers and have reanalyzed data set that now show that nasal epithelial responses to CoV are similar to those induced by other respiratory viruses. Data regarding ACE2 and TMPRSS2 expression i.e. ACE2 suppressed in TMPRSS2 induced and associated with TH2 gene networks are strong and derived from both in vitro, experimental, and clinical data. Present QTL data will be of interest once COVID-19 genome-wide data, susceptibility, and severity data are available for integration from the pandemic. It remains hypothetical whether levels of ACE2 and TMPRSS2 are related to susceptibility or severity of COVID-19 infection.

The text has been modified and statements modified appropriately to reflect the importance of data, however, several statements may need further modification.

Figure 1: The legend might better explain the experiments (a.) what is the n=?; how were data normalized? For example to what?

Pg. 19: Single-cell RNA seq analysis of n=1 asthmatic is not sufficient and merits a larger or simply reference the published work suffices to support their conclusions regarding bulk seq RNA data.

Figure 5: Are predicted QTLs in coding or predicted regulatory region of the ACE2 or TMPRSS2 genes, for example, does encode or chip seq database support the regulatory functions of the QTLs?

Pg. 21: Provisionally suggests that T2 inflammation may predispose "is overly speculative" since it is unclear whether levels of either protein influence susceptibility to infection.

Pg. 23: it is completely speculative and over-reaching to state that "targeted blockade of this interferon effect could control COVID-19".

Pg. 24: The statement that their data support the investigation of IL-6 blockade might be modified since there is long-standing justification to block IL-6 for therapeutic purposes.

Reviewer #1

I appreciate the authors' thorough response to the review and think that the revision to include comparisons to a broader "other respiratory virus" group significantly enhances the analysis. The fact that it is "negative data" in no way detracts from its value to the field--I think this will be a very useful resource.

The other edits address my primary concerns.

Response: We thank the reviewer for their careful critique of the manuscript, which has greatly improved its quality.

Reviewer #2

The revised manuscript is improved, though not sufficiently. The key points regarding have not been fully addressed:

1. transcriptional comparison of "all coronaviruses" vs. rhinoviruses was largely dismissed. Viruses are unique and elicit specific responses. Pooling of all coronaviruses really misses the mark.

Response: As we indicated in our first response, due to the heterogeneity of our sample, we lack the statistical power to evaluate each of the four human coronavirus species individually. Of the samples with sufficient levels (>1000 reads) of viral reads to suggest a potentially robust infection, 4 were infected with OC43, 3 with HKU1, and 1 each with 229E and NL63. In their 2016 paper in *RNA*, Schurch, *et al* report that to detect medium to highly differential genes 6 replicates are needed for each condition, and to increase the sensitivity of the analysis sufficiently to capture smaller changes 12 replicates are better¹. With this benchmark, 1-4 samples are simply insufficient for analyzing individual coronavirus species separately.

Moreover, our primary goal was not to characterize the response to each coronavirus species, but rather to extract generalizable features of coronavirus responses that may also be applicable to SARS-CoV-2 and differentiate this species as a whole from others. Although this analysis did not yield positive results we still believe these negative results are important to the field as noted by another reviewer.

2. there are no protein data in the original manuscript and none in the revision. This reduces the impact of the paper.

Response: We agree with the reviewer that protein data would enhance the paper. Although *in vivo* protein data is not available for the GALA cohort used in the population-based analyses, we were able to leverage available stimulated (IL-13, Virus, Notch Inhibitor), *in vitro*, mucociliary airway epithelial cultures and *in vivo* tracheal airway epithelial sections to extend some of our findings to the protein level for ACE2. We were unable to successfully stain for TMPRSS2 using available antibodies. Excitingly, through this staining we were able to extend our major findings that T2 and viral stimulation of the epithelium impact ACE2 from mRNA to

the protein level. The findings garnered from our protein work are detailed below and presented in the new paper section “ACE2 protein expression in the airway epithelium is regulated by T2 inflammation and viral infection” and the new Figure 5.

Findings from our Protein Staining

1. We find prominent ACE2 protein on the apical surface of the epithelium, colocalizing mostly with ciliated cells.
2. Acute IL-13 treatment of the nasal airway epithelium down regulates apical ACE2 protein levels, while HRV infection upregulates apical ACE2 protein levels.
3. Differentiation of basal airway epithelial cells into a mucociliary epithelium in the presence of a gamma-secretase inhibitor (DAPT), which drives near uniform differentiation of ciliated cells, greatly increases apical ACE2 protein levels. In contrast, differentiation of basal airway epithelial cells into a mucociliary epithelium in the presence of IL-13, which drives near uniform differentiation of mucus cells, greatly decreases apical ACE2 protein levels.

Reviewer #3

The authors have carefully considered the comments from the reviewers and have reanalyzed data set that now show that nasal epithelial responses to CoV are similar to those induced by other respiratory viruses. Data regarding ACE2 and TMPRSS2 expression i.e. ACE2 suppressed in TMPRSS2 induced and associated with TH2 gene networks are strong and derived from both in vitro, experimental, and clinical data. Present QTL data will be of interest once COVID-19 genome-wide data, susceptibility, and severity data are available for integration from the pandemic. It remains hypothetical whether levels of ACE2 and TMPRSS2 are related to susceptibility or severity of COVID-19 infection.

The text has been modified and statements modified appropriately to reflect the importance of data, however, several statements may need further modification.

Figure 1: The legend might better explain the experiments (a.) what is the n=?; how were data normalized? For example, to what?

Response: We agree the legends could be more informative. We have modified the legend text to include the number of cells and the normalization method used in the data presentation.

Pg. 19: Single-cell RNA seq analysis of n=1 asthmatic is not sufficient and merits a larger or simply reference the published work suffices to support their conclusions regarding bulk seq RNA data.

Response: This question had been brought up by reviewer 2 during the first revision. We use the information from the nasal brushing single cell RNA-seq analysis to simply establish cell

types expressing *ACE2* and *TMPRSS2* in the asthmatic nasal epithelium, not to draw any broader conclusions. Moreover, our results are in line with two other more extensive scRNA-seq papers on these genes which we referenced in the discussion. Finally, we explicitly state the limitation of using one donor, See page 21, line 461-463.

Figure 5: Are predicted QTLs in coding or predicted regulatory region of the *ACE2* or *TMPRSS2* genes, for example, does encode or chip seq database support the regulatory functions of the QTLs?

Response:

snp	chr	pos_hg38	DNase	ChIP	PWM matched	Footprint matched	RegulomeDB	
							probability	ranking
rs181603331	X	15540999	0	0	0	0	0.18	7
rs2838057	21	41550890	1	0	0	0	0.35	5
rs1475908	21	41560130	1	1	0	0	0.61	4
rs74659079	21	41778415	1	1	1	1	0.96	2a

None of the eQTL variants are located in the gene body of either *ACE2* or *TMPRSS2*. We use regulomeDB² to query the eQTL variants for evidence of regulatory sites from ENCODE as presented in the table above. The RegulomeDB probability score represents a model integrating functional genomics features and is ranging from 0 to 1, with 1 being most likely to be a regulatory variant. Based on RegulomeDB annotation, 3 of the 4 eQTL variants have reasonable regulatory variant probabilities.

Pg. 21: Provisionally suggests that T2 inflammation may predispose “is overly speculative” since it is unclear whether levels of either protein influence susceptibility to infection.

Response: We agree this point is overly speculative and we have removed from the manuscript. Additionally, we have further modified the statement following the referenced one, to raise the possibility that T2 inflammation may not have an effect on COVID-19 disease.

“Both *in vitro* experiments examining IL-13 effects on SARS-CoV-2 infection and empirical data on COVID-19 outcomes among T2-high and T2-low patients will be needed to determine whether this common airway inflammatory endotype ultimately protects against, exacerbates, or has no effect on COVID-19 illness.”

Pg. 23: it is completely speculative and over-reaching to state that “targeted blockade of this interferon effect could control COVID-19”.

Response: Although, we believe our speculation is logical and well-grounded we will remove this statement and let a recently generated piece of data make a similar point. Please see the new statement below referencing this work.

“Supporting a possible role for interferon responses in COVID-19 outcomes, a recent single cell RNA-seq study found increased type I interferon response was a defining feature of patients with a severe vs. mild COVID-19 illness.”³

Pg. 24: The statement that their data support the investigation of IL-6 blockade might be modified since there is long-standing justification to block IL-6 for therapeutic purposes.

Response: Our statement has been modified to more clearly address that our data is supplemental to existing data:

“Notably, we found that IL-6 was predicted to regulate airway responses to CoV and was itself upregulated in the airway with these infections. These data provide additional support for the ongoing investigation of tocilizumab (IL-6R blocking antibody) for the treatment of COVID-19 illnesses.”

References:

1. Schurch, N.J. *et al.* How many biological replicates are needed in an RNA-seq experiment and which differential expression tool should you use? *RNA* **22**, 839-851 (2016).
2. Dong, S. & Boyle, A.P. Predicting functional variants in enhancer and promoter elements using RegulomeDB. *Hum Mutat* **40**, 1292-1298 (2019).
3. Lee, J.S. *et al.* Immunophenotyping of COVID-19 and influenza highlights the role of type I interferons in development of severe COVID-19. *Sci Immunol* **5** (2020).